# QUTCC 🤗: Quantile Uncertainty Training and Conformal Calibration for Imaging Inverse Problems

## Abstract

Deep learning models often hallucinate, producing realistic artifacts that are not truly present in the sample. This can have serious consequences in scientific and medical inverse problems, where accuracy is more important than perceptual quality. Uncertainty quantification techniques, such as conformal prediction, can pinpoint outliers and provide guarantees for image regression tasks, improving reliability. However, existing methods predict fixed quantiles and utilize a linear constant scaling factor to calibrate uncertainty bounds, resulting in larger, less informative bounds. We propose *QUTCC*, a quantile uncertainty training and calibration technique that enables nonlinear, non-uniform scaling of quantile predictions to enable tighter uncertainty estimates. Using a U-Net architecture with a quantile embedding, QUTCC can predict the full conditional distribution of quantiles for each image. After conformal calibration, QUTCC can predict pixel-wise uncertainty intervals that satisfy coverage guarantees and also estimate a pixel-wise conditional probability density function. We evaluate our method on image denoising, quantitative phase imaging, and compressive MRI reconstruction. Our method successfully pinpoints hallucinations in image estimates and consistently achieves tighter uncertainty intervals than prior methods while maintaining the same statistical coverage.

## 1 Introduction

In recent years, deep learning models have become the dominant approach across many inverse problems, favored for their ability to learn powerful and complex priors from an abundance of data (Ongie et al., 2020; Alshardan et al., 2024; Barbastathis et al., 2019; Xue et al., 2019). However, these models are generally limited in their ability to represent uncertainty in their predictions, which has been a significant barrier to their use in scientific and medical applications, where identifying observations as out-of-distribution is crucial and the consequences of model hallucination can be severe (Begoli et al., 2019). While significant progress has been made in estimating uncertainty in deep learning, many methods require substantial computational demands, incorporate strong prior data assumptions, or may not provide formal statistical guarantees (Psaros et al., 2023; Gal & Ghahramani, 2016; Gal et al., 2017; Lakshminarayanan et al., 2017; Sun et al., 2024). These limitations have motivated research into conformal prediction methods for uncertainty quantification, which aim to overcome these challenges (Angelopoulos & Bates, 2021; Angelopoulos et al., 2021; Dewolf et al., 2023; Barber et al., 2023; Romano et al., 2019).

Specifically, recent methods apply the statistical rigor of conformal prediction to inverse problems and image regression tasks through multi-dimensional conformalized quantile regression (Angelopoulos et al., 2022b; Romano et al., 2019; Koenker & Bassett Jr, 1978). This is typically done by modifying a neural network to output two additional predictions: a lower-bound and an upper-bound image, which together define a confidence interval for each pixel. A calibration step is then used to scale these bounds so that the interval captures the true intensity values with the desired coverage, resulting in pixel-wise intervals that are statistically guaranteed to contain the true values with a user-specified confidence level (Angelopoulos et al., 2022b). Multi-dimensional conformalized quantile regression is simple to implement and computationally inexpensive, unlike ensemble methods, which often require

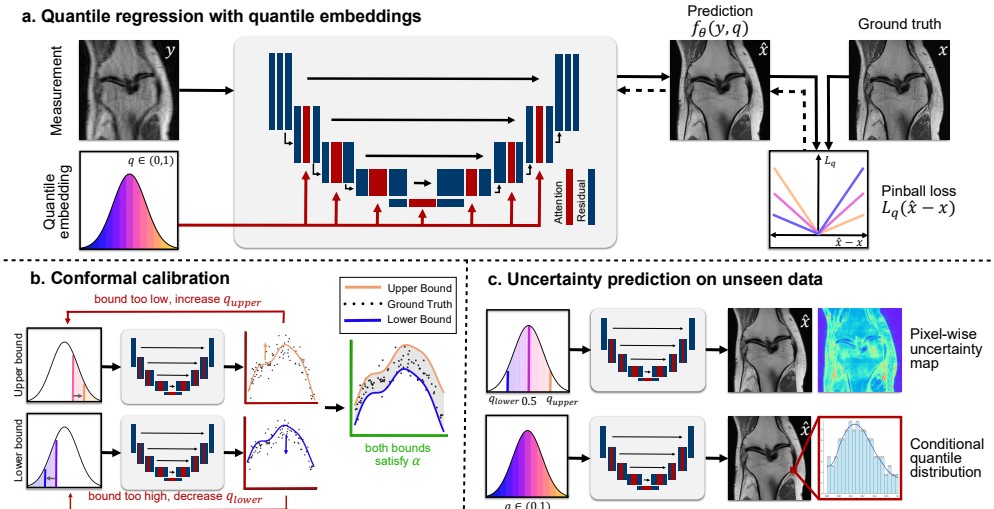

Figure 1: **QUTCC Overview. a) Quantile Regression with Quantile Embedding:** During training, a neural network with a quantile embedding predicts an image as a function of the measurement and quantile, $q$. The quantile embedding is randomly sampled ($q \in (0,1)$) and the value of $q$ determines the asymmetry of the pinball loss, enabling the model to learn a range of conditional quantiles. **b) Conformal Calibration:** During calibration, the predictive bounds ($q_{\text{lower}}, q_{\text{upper}}$) are iteratively adjusted on a held-out dataset to satisfy the desired miscoverage level $\alpha$. **c) Uncertainty Prediction on Unseen Data:** At test time, the model can be queried to predict the mean image, a pixel-wise uncertainty map, or a conditional probability density function at each pixel.

training multiple models, while still providing formal coverage guarantees that are independent of the model choice or data distribution. This makes it particularly well-suited for scientific and medical imaging, which often has small datasets, and adaptive imaging, where inference speed is crucial (Ye et al., 2025). However, prior methods use a linear pixel-wise scaling during conformal calibration to approximate a complex, nonlinear distribution, which leads to overly loose prediction intervals (Angelopoulos et al., 2022b). Furthermore, existing methods learn fixed quantiles, which carry the risk of quantile crossing (He, 1997).

To address these challenges, we present **Quantile Uncertainty Training and Conformal Calibration (QUTCC)**, a novel method for simultaneous quantile prediction and conformal calibration that enables efficient and accurate uncertainty quantification for imaging inverse problems (Fig 1). Building on past work in multi-quantile estimation (Rodrigues & Pereira, 2020; Tagasovska & Lopez-Paz, 2019), QUTCC uses a single neural network to estimate a distribution of quantiles. During the conformal calibration step, QUTCC applies a non-uniform, nonlinear scaling to the uncertainty bounds, compared to constant scaling used by prior methods. This results in smaller and potentially more informative uncertainty intervals. Additionally, because all quantiles are learned during training, QUTCC can query the full quantile range at inference time to construct a pixel-wise estimate of the underlying probability distribution. We evaluate QUTCC on five imaging inverse problems and show that QUTCC can highlight regions of hallucination and maintain smaller uncertainty interval sizes than prior methods, while satisfying the same statistical guarantees. Additionally, we show that QUTCC effectively approximates pixel-wise probability density functions, providing a richer representation of uncertainty that could enable more informed downstream decision-making in scientific and medical imaging applications.

## 2 RELATED WORK

### 2.1 CONFORMAL PREDICTION

Conformal prediction is a robust statistical technique that constructs predictive intervals with a formal guarantee of statistical coverage (Angelopoulos & Bates, 2021; Correia et al., 2024; Romano et al.,

2019). In general, conformal methods take heuristic or model-derived uncertainty estimates and refine them into statistically valid predictive intervals by leveraging a held-out calibration dataset. These intervals are constructed to satisfy a finite-sample marginal coverage guarantee at a user-specified level $(1 - \alpha)$, meaning that, with high probability, the true response lies within the predicted interval at least $(1 - \alpha)\%$ of the time. For example, setting $\alpha = 0.1$ yields prediction intervals that are guaranteed to contain the ground truth in at least $90\%$ of test cases. Conformal prediction has become increasingly popular due to its simplicity, speed, formal guarantees, and lack of assumptions on model or data distribution (Angelopoulos et al., 2023). It has been applied to a variety of areas, including classification Angelopoulos et al. (2020); Ding et al. (2023); Angelopoulos et al. (2021), language modeling (Quach et al., 2023; Campos et al., 2024), robotics (Lindemann et al., 2023; Lekeufack et al., 2024), protein design (Fannjiang et al., 2022), and time series estimation (Xu & Xie, 2023; Zaffran et al., 2022). Recently, conformal prediction has been proposed for image-to-image regression tasks to enable pixel-wise uncertainty prediction (Angelopoulos et al., 2022b), and thereafter applied to adaptive-microscopy (Ye et al., 2025). These methods alter a neural network to predict fixed upper and lower quantile estimates, in addition to the mean image, then uniformly scale these predictions by a constant factor using conformal risk control to ensure valid statistical coverage at test time (Angelopoulos et al., 2022b). Instead of predicting fixed quantiles and uniformly scaling them during conformal calibration, we use one network to predict all possible conditional quantiles. This enables non-uniform, asymmetric scaling during conformal calibration and leads to tighter uncertainty intervals for image regression tasks and imaging inverse problems.

## 2.2 QUANTILE REGRESSION

Quantile regression is a general approach to estimate the conditional quantiles of a target distribution rather than the mean of a response variable (Koenker & Bassett Jr, 1978; Koenker & Hallock, 2001). This is often accomplished by leveraging an asymmetric loss function, called pinball loss (Fig. 1a , Eq. 3), tailored to the specified quantile level (Steinwart & Christmann, 2011). The estimated intervals obtained by quantile regression do not have formal guarantees on their own, but can be paired with conformal prediction to obtain coverage guarantees (Romano et al., 2019). Learning quantiles during neural network training can improve predictive performance by introducing a regularization effect, while enabling uncertainty estimation (Rodrigues & Pereira, 2020). One limitation of quantile-based methods is the tendency for 'quantile crossing' to occur, a phenomenon wherein quantiles trained independently violate their natural ordering, resulting in lower quantiles exceeding higher ones (Das et al., 2019). Training a single network to simultaneously predict multiple quantiles, called 'simultaneous quantile prediction' can mitigate this issue, while also enabling the estimation of the entire conditional distribution (Sangnier et al., 2016; Liu & Wu, 2011; Tagasovska & Lopez-Paz, 2019; Rodrigues & Pereira, 2020). In our work, we leverage a single-network with shared parameters for simultaneous quantile prediction. We embed the quantile level as an explicit input parameter into a U-Net, which is well-suited for a variety of image regression and imaging inverse tasks. Embedding a notion of quantiles into deep learning architectures has been explored in reinforcement learning and generative modeling Dabney et al. (2018); Ostrovski et al. (2018), but its application to predicting uncertainty bounds for image regression tasks remains largely unexplored. Furthermore, we pair our network with conformal prediction to achieve coverage guarantees. We show that our network mitigates the issue of quantile crossing while also maintaining overall prediction accuracy. We are the first to demonstrate that a single network trained for simultaneous quantile prediction can predict conformally calibrated uncertainty intervals for imaging inverse problems.

## 2.3 PREDICTING TIGHTER OR MORE INTERPRETABLE BOUNDS

Achieving smaller interval lengths without sacrificing coverage guarantees reflects increased confidence in the model's predictions, leading to more precise and reliable uncertainty quantification. Producing tighter bounds is a common objective across uncertainty estimation methods, not just conformal prediction (Xie et al., 2024). Several approaches have been proposed to enhance conformal prediction by targeting user-specified properties such as reduced interval length or improved conditional coverage (Xie et al., 2024; Chung et al., 2021); however, to date, none of these techniques have been applied to imaging tasks. On the other hand, several methods aim to improve the interpretability of uncertainty prediction for imaging tasks by moving away from per-pixel uncertainty estimates. These methods leverage principal components, posterior projected distribution, and spatial/topological relationships (Nehme et al., 2023; Yair et al., 2024; Belhasin et al., 2023; Gupta et al., 2023) to

predict uncertainty in a more interpretable way. Furthermore, studies in multi-hypothesis uncertainty estimation have investigated multi-head and mixture-based networks that predict multiple candidate hypotheses to explore the space of potential solutions instead of presenting a single candidate reconstruction (Rupprecht et al., 2017; Ilg et al., 2018; Nehme et al., 2024). However, without incorporating conformal prediction, these methods lack statistical guarantees. Several methods pair inverse problems with downstream tasks, such as classification, to estimate the uncertainty in a more interpretable way (Cheung et al., 2024; Wen et al., 2024), and others represent uncertainty in a semantically-meaningful latent space (Sankaranarayanan et al., 2022). While these methods are promising, they are less general and often tied to a specific application. Additional work have also investigated risk controlling prediction sets for image regression tasks in generative and medical tasks (Fischer et al., 2024; Teneggi et al., 2023). We present a more general method that can predict uncertainty for any imaging inverse problem while achieving smaller uncertainty interval lengths than previous image-to-image regression methods.

## 3 METHODOLOGY

### 3.1 PROBLEM OVERVIEW

We focus on predicting a image, $\mathbf{x} \in \mathbb{R}^{d_x}$, from a measurement, $\mathbf{y} \in \mathbb{R}^{d_y}$, given some imaging operator/degradation and noise $n$, $\mathbf{y} = A(\mathbf{x}) + n$. This general framework applies to a wide range of imaging problems, including denoising, deblurring, super-resolution, magnetic resonance imaging (MRI), computer tomography (CT), and phase retrieval. This problem is often ill-conditioned or underdetermined, and it may be impossible to perfectly infer $\mathbf{x}$ from $\mathbf{y}$. We assume that we have access to matched input, output pairs, $\mathcal{D}_t = \{(\mathbf{x}_i, \mathbf{y}_i)\}_{i=1}^{N_t}$, that are randomly sampled from the unknown joint distribution $p(\mathbf{x}, \mathbf{y})$.

On a high level, our goal is to predict an image $\hat{\mathbf{x}}$ and its uncertainty $\hat{\mathbf{u}}$, given an unseen measurement, $\mathbf{y}$. To do this, we train a neural network, $f_\theta(\mathbf{y}, q)$[1], with parameters $\theta$ that can predict any conditional quantile, $q$, of the joint distribution. The image estimate is obtained by querying the network at $q = 0.5$, which gives us the median image:

$$\hat{\mathbf{x}} = f_\theta(\mathbf{y}, q = 0.5). \tag{1}$$

The uncertainty interval is obtained by querying the network at a high quantile value, $q_{\text{upper}}$, and a low quantile value, $q_{\text{lower}}$. The upper quantile prediction serves as the upper bound of the uncertainty interval, and the lower quantile serves as the lower bound. These two quantities can be subtracted to get a notion of pixel-wise uncertainty: $\hat{\mathbf{u}} = f_\theta(\mathbf{y}, q_{\text{upper}}) - f_\theta(\mathbf{y}, q_{\text{lower}})$.

During conformal calibration, we use a small calibration dataset, $\mathcal{D}_c = \{(\mathbf{x}_i, \mathbf{y}_i)\}_{i=1}^{N_c}$, to search over different values of the input parameters $q_{upper}$ and $q_{lower}$ until we reach the desired coverage. That is, the constructed interval, $\mathcal{C} \in [f_\theta(\mathbf{y}, q_{\text{lower}}), f_\theta(\mathbf{y}, q_{\text{upper}})]$, contains at least $1 - \alpha$ of the ground truth pixels (Angelopoulos et al., 2022a). Formally, this means that:

$$\mathbb{E}\Big[\mathbf{x}_{\text{test}}[k] \in C(\mathbf{y}_{\text{test}}[k])\Big] \geq 1 - \alpha, \quad \forall k \in \{1, \ldots, K\}, \tag{2}$$

where $\mathbf{x}_{\text{test}}$ and $\mathbf{y}_{\text{test}}$ are unseen test images from the same distribution as the calibration set, where $k$ is the pixel index with $K$ total pixels in the image. After calibration, the network can provide a pixel-wise uncertainty map for new, unseen data or can be queried to predict a conditional distribution for any pixel in the image. Our method QUTCC, is summarized in Fig. 1. We elaborate on our network architecture, training procedure, and the conformal calibration step below.

### 3.2 NETWORK ARCHITECTURE

Our proposed network, $f_\theta(\mathbf{y}, q)$, takes the parameter $q$ as an input to predict the conditional quantile. To do this in practice, we propose to use an attention U-Net (Ronneberger et al., 2015; Oktay et al., 2018) with a quantile-embedding to condition the U-Net on a given $q$. This is inspired by

---

[1]Please note that we adopt the notation $\hat{x} = f(y)$, which is commonly used in the field of inverse problems instead of $\hat{y} = f(x)$, which is more common in the machine learning literature.

the architecture of U-Nets used for diffusion models, which include a time-embedding (Ho et al., 2020; Zhang et al., 2023). Specifically, we encode randomly sampled quantiles ($q \in (0,1)$) as high-dimensional vectors during training, allowing the network to learn the distribution of the target variable for each specified quantile. While prior work has used quantile embeddings to condition deep networks (Dabney et al., 2018; Ostrovski et al., 2018), our architecture integrates these embeddings into a self-attention U-Net intended for image regression and uncertainty quantification. This allows us to query an image prediction at any quantile during a forward pass. The neural network weights are shared across different quantile predictions, limiting quantile crossing. The proposed network is shown in Fig. 1, and the full architecture and training details are described in the Supplement.

## 3.3 SIMULTANEOUS QUANTILE REGRESSION

In order to train our neural network to predict an arbitrary quantile image, we use pinball loss ($L_q$), an asymmetric loss function commonly used in quantile regression (Eq. 3), where $\hat{x}$ denotes the predicted value, $x$ represents the ground truth, and $q \in (0,1)$ is the quantile of interest:

$$L_q(x, \hat{x}) = \begin{cases} q \cdot |x - \hat{x}| & \text{if } x - \hat{x} \geq 0 \\ (1 - q) \cdot |x - \hat{x}| & \text{otherwise.} \end{cases} \tag{3}$$

When $q \neq 0.5$, the pinball loss introduces asymmetry by penalizing overestimates and underestimates unequally. Specifically, when $q > 0.5$, the loss assigns a greater penalty to underestimations (i.e., when $\hat{x} < x$), encouraging the model to predict higher values. Conversely, for $q < 0.5$, overestimations (i.e., when $\hat{x} > x$) incur a larger penalty, biasing predictions downward. This asymmetry enables the model to learn conditional quantiles of the target distribution, in contrast to losses like mean squared error (MSE), which are symmetric and designed to estimate the conditional mean. At each training step, the quantile parameter, $q$ is randomly sampled and used to both condition the network and as an input to the loss function. This allows the model to learn the full conditional quantile function, rather than a discrete, fixed quantile value as in prior image-to-image regression methods (Angelopoulos et al., 2022b; Ye et al., 2025). The total loss is given by:

$$\mathcal{L}_{\text{total}}(\theta) = \mathbb{E}_{[(x,y) \sim D_{\text{t}}, q \sim \mathcal{U}(0,1)]} \Big[ \mathcal{L}_q \big( x, f_\theta(y, q) \big) \Big], \tag{4}$$

where $f_\theta$ is a neural network with parameters $\theta$, and $f_\theta(y_i, q)$ is the output of the neural network given an input measurement, $y_i$ and quantile value $q$. The loss in Eq. equation 4 is defined as an expectation over both the data distribution $(x, y) \sim D_{\text{t}}$ and quantile levels $q \sim \mathcal{U}(0,1)$, explicitly reflecting the uniform sampling used during training. This loss is minimized with the Adam optimizer (Kingma & Ba, 2014) using backpropagation.

## 3.4 CONFORMAL CALIBRATION

After training, the neural network can be queried to obtain the uncertainty interval predictions. However, these predictions may not be valid. To ensure the statistical coverage in Eq. 2, a conformal calibration step is necessary. Following the procedure in (Angelopoulos et al., 2022b), we use a small, separate calibration dataset, $\mathcal{D}_c = \{(\mathbf{x}_i, \mathbf{y}_i)\}_{i=1}^{N_c}$ to adjust $q_{\text{lower}}$ and $q_{\text{upper}}$ until the desired coverage, $1 - \alpha$, is reached. Our model's risk over the calibration dataset is the number of miscovered pixels for each image normalized by the total number of pixels, $K$, and the size of the calibration dataset, $N_c$:

$$\hat{R}(q_{\text{lower}}, q_{\text{upper}}) = \frac{1}{N_c} \sum_{i=1}^{N} \frac{1}{K} \sum_{k=1}^{K} \mathbb{1}\{\mathbf{x}_i(k) \notin \mathcal{C}(\mathbf{y}_i(k), q_{\text{lower}}, q_{\text{upper}})\}, \tag{5}$$

where $k$ is the index of the pixels in the image. This measures the miscoverage as a function of $q_{\text{lower}}$ and $q_{\text{upper}}$. We can decompose this total risk as a function of the miscoverage from the lower quantile (the number of ground truth pixels that are lower than the lower quantile) and the miscoverage from the upper bound (the number of ground truth pixels that are higher than the upper quantile):

$$\hat{R}(q_{\text{lower}}, q_{\text{upper}}) = \frac{1}{N_c} \sum_{i=1}^{N} \frac{1}{K} \sum_{k=1}^{K} \left[ \mathbb{1}\{\mathbf{x}_i(k) < f_\theta(\mathbf{y}_i, q_{\text{lower}})\} + \mathbb{1}\{\mathbf{x}_i(k) > f_\theta(\mathbf{y}_i, q_{\text{upper}})\} \right]. \tag{6}$$

During conformal calibration, we calibrate the lower and upper quantile bounds independently. To satisfy a target total miscoverage rate of $\alpha$, the calibration process allocates half of this error budget to each bound. That is, the lower and upper bounds are each adjusted to capture violations at a rate no greater than $\alpha/2$. Thus, if $\mathbf{x}(k)$ denotes the ground truth at pixel $k$ and $[\hat{\mathbf{x}}_{lower}(k), \hat{\mathbf{x}}_{upper}(k)]$ the predicted interval, the goal is to ensure:

$$\mathbb{P}(\mathbf{x}(k) < \hat{\mathbf{x}}_{\text{lower}}(k)) \leq \frac{\alpha}{2} \quad \text{and} \quad \mathbb{P}(\mathbf{x}(k) > \hat{\mathbf{x}}_{\text{upper}}(k)) \leq \frac{\alpha}{2}, \tag{7}$$

which implies:

$$\mathbb{P}\big(\hat{\mathbf{x}}_{\text{lower}}(k) \leq \mathbf{x} \leq \hat{\mathbf{x}}_{\text{upper}}(k)\big) \geq 1 - \alpha. \tag{8}$$

By allowing the quantile bounds to vary independently and adaptively during calibration, our method supports non-uniform scaling as a function of image characteristics. Rather than scaling the quantile predictions by a constant, linear factor, as proposed in prior work (Angelopoulos et al., 2022b), our quantile predictions can be scaled in a non-uniform, non-linear way as a function of the neural network, $f_\theta(\mathbf{y}, q_{\text{lower}}, q_{\text{upper}})$. In practice, this can result in smaller uncertainty intervals.

Pseudocode for this calibration process is provided in **Algorithm 1**. At each step, we compute the miscoverage from the quantile upper and lower bounds over the entire calibration dataset. If the violation rate for a bound exceeds the adjusted $\alpha$, we relax the corresponding bound; otherwise, we tighten it. This process proceeds via a binary search over the quantile space until the desired coverage is reached. Conducting a binary search over the quantile space assumes that the learned quantile function is monotonic. In practice, this function is mostly monotonic with a few rare violations, which occur in background regions (See supplement Tbl. 3a). Note that we adjust the error rate, $\alpha' \leftarrow \alpha - \frac{1-\alpha}{N_c}$ to account for the finite calibration dataset size (Angelopoulos & Bates, 2021; Vovk, 2012). At the end of this procedure, we can obtain a constructed interval $\mathcal{C} \in [f_\theta(\mathbf{y}, q_{\text{lower}}^*), f_\theta(\mathbf{y}, q_{\text{upper}}^*)]$ that satisfies Eq. 2.

---

**Algorithm 1** Calibrating Quantile Bounds $q_{\text{lower}}, q_{\text{upper}}$

---

1: Compute adjusted error: $\alpha' \leftarrow \alpha - \frac{1-\alpha}{N_c}$
2: Initialize bounds: $q_{\text{lower}} \leftarrow \alpha'$, $q_{\text{upper}} \leftarrow 1 - \alpha'$
3: Define step size $\Delta q$ for bound updates
4: **while** $R(q_{\text{lower}}, q_{\text{upper}}) > \alpha'$ **do**
5: $\quad (r_{\text{lower}}, r_{\text{upper}}) \leftarrow (R_{\text{lower}}(q_{\text{lower}}), R_{\text{upper}}(q_{\text{upper}}))$
6: $\quad$ Update $q_{\text{lower}} \leftarrow q_{\text{lower}} \pm \Delta q$ $\qquad\qquad\qquad$ ▷ $-\Delta q$ if $r_{\text{lower}} \leq \alpha'/2$, else $+\Delta q$
7: $\quad$ Update $q_{\text{upper}} \leftarrow q_{\text{upper}} \pm \Delta q$ $\qquad\qquad\qquad$ ▷ $+\Delta q$ if $r_{\text{upper}} \leq \alpha'/2$, else $-\Delta q$
8: **return** $(q_{\text{lower}}^*, q_{\text{upper}}^*)$

---

### 3.4.1 ESTIMATING THE CONDITIONAL DISTRIBUTION

Since our network is trained to predict the full quantile function, rather than a single fixed quantile, we can recover an estimate of the entire quantile function $\hat{Q}_k(q)$ at each pixel k. This is accomplished by querying the network over a range of quantile levels $q \in (0, 1)$. Assuming that the pixel-wise distribution is continuous, the pixel-wise CDF $F_k(x)$ is strictly increasing and the true quantile function $Q_k(q)$ is differentiable, the following relationship holds:

$$f_k(Q_k(q)) = \frac{1}{Q_k'(q)} \tag{9}$$

We can obtain an estimate of the derivative $\hat{Q}_k'(q)$ of the estimated quantile function through numerical approximation using finite differences. This allows us to obtain an estimate of the PDF $f_k(x)$ for pixel intensity $x := \hat{Q}_k(q)$, by the following formula:

$$\hat{f}_k(x) = \frac{1}{\hat{Q}_k'(q)}. \tag{10}$$

In practice, we query our neural network at different quantile levels $\{q_i\}_{i \in [n]}$ to obtain predictions $x_i := \hat{Q}_k(q_i)$, which in turn can be used to compute $\hat{Q}_k'(q_i) = \frac{q_{i+1} - q_{i-1}}{x_{i+1} - x_{i-1}}$. We finally obtain estimates

of the PDF at points $x_1, \cdots, x_n$ by taking the inverse of $\hat{Q}'_k(q_i)$ for all $i \in [n]$. This approach not only enables accurate pixel-wise PDF estimation but, when combined with conformal calibration of multiple bounds, also provides statistically guaranteed coverage of the estimated density.

To 'conformalize' the pixel-wise PDF estimation, we first specify the desired coverage level (e.g., for 90% coverage, we initialize with quantiles $0.05$ and $0.95$; for 60%, with $0.20$ and $0.80$). These initial bounds are then calibrated using Algorithm 1 to guarantee the target coverage. By performing this calibration for multiple quantile levels $\{q_i\}_{i=1}^n$ and their associated coverage rates $\{1 - \alpha_i\}_{i=1}^n$, we obtain a set of conformally corrected quantiles,

$$\{\hat{Q}_k^{\text{conf}}(q_i)\}_{i=1}^n, \tag{11}$$

from which the pixel-wise PDF is reconstructed via finite differences:

$$\hat{p}_k^{\text{conf}}\big(\hat{Q}_k^{\text{conf}}(q_i)\big) = \left( \frac{\hat{Q}_k^{\text{conf}}(q_{i+1}) - \hat{Q}_k^{\text{conf}}(q_{i-1})}{q_{i+1} - q_{i-1}} \right)^{-1}. \tag{12}$$

By calibrating multiple quantile levels with their corresponding coverage rates, we construct a PDF that provides reliable and conformally calibrated estimates.

## 4 RESULTS

To evaluate our proposed approach, we fit and calibrate a separate model $f_\theta$ to five separate imaging inverse problems: accelerated MRI (Zbontar et al., 2018), quantitative phase imaging (QPI) (Pinkard et al., 2024), and denoising under real-noise, synthetic Poisson, and Gaussian noise (Zhang et al., 2019). We compare against Im2Im-UQ, which is the leading conformal prediction approach for image-to-image regression. To ensure that our performance improvements come from our uncertainty quantification technique and not network improvements, we upgrade Im2Im-UQ to use the same architecture and depth as QUTCC, which we call Im2Im-Deep. In all cases, we set $\alpha = 0.1$ during conformal calibration. Full training details are provided in the supplement. We evaluate predicted interval lengths, empirical risk, and model performance. We also visualize uncertainty and highlight the model's ability to identify hallucinations, which are realistic features not present in the ground truth. Finally, we show that QUTCC infers pixel-wise PDFs without relying on prior distributional assumptions.

### 4.1 UNCERTAINTY INTERVAL LENGTH AND RISK

We compare the predicted uncertainty interval lengths and total risk for each imaging modality in Fig. 2 and Table 1. QUTCC consistently produces smaller prediction intervals than Im2Im-Deep across all five modalities, while keeping the total risk under 0.1. By achieving smaller interval lengths while exhibiting comparable risk, QUTCC demonstrates that its uncertainty quantification is both more precise and well-calibrated without sacrificing coverage.

In Figure 3, we compare QUTCC and Im2Im-Deep for Gaussian denoising with progressively increasing noise, plotting the average uncertainty interval lengths as a function of pixel intensity. When the noise variance is lower, QUTCC produces smaller interval lengths that better match the noise level compared to Im2Im-Deep. In addition, QUTCC consistently estimates narrower uncertainty intervals in regions of high signal intensity. We further demonstrate this in Suppl. Fig. 6, where uncertainty interval sizes are stratified by pixel intensity. Across Gaussian, Poisson, and real-noise settings, QUTCC consistently yields smaller intervals in regions of higher pixel intensity.

| Metric | Method | MRI | QPI | Gaussian | Poisson | Real Noise |
|---|---|---|---|---|---|---|
| Interval Length | Im2Im-Deep | $0.109 \pm 0.056$ | $0.065 \pm 0.014$ | $0.063 \pm 0.049$ | $0.047 \pm 0.038$ | $0.038 \pm 0.045$ |
| | QUTCC | $\mathbf{0.108 \pm 0.057}$ | $\mathbf{0.062 \pm 0.014}$ | $\mathbf{0.059 \pm 0.048}$ | $\mathbf{0.040 \pm 0.029}$ | $\mathbf{0.036 \pm 0.035}$ |
| Total-Risk | Im2Im-Deep | $0.099 \pm 0.047$ | $0.098 \pm 0.099$ | $0.094 \pm 0.065$ | $0.049 \pm 0.042$ | $0.096 \pm 0.040$ |
| | QUTCC | $0.097 \pm 0.031$ | $0.100 \pm 0.102$ | $0.090 \pm 0.046$ | $0.093 \pm 0.091$ | $0.098 \pm 0.029$ |

Table 1: Interval Length and Total Risk

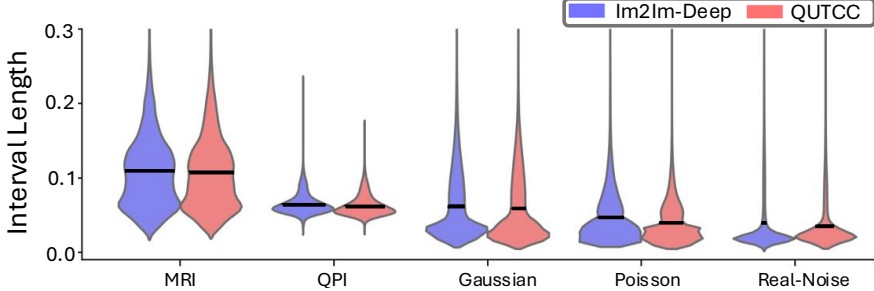

Figure 2: **QUTCC exhibits smaller uncertainty interval sizes:** We compare the predictive interval sizes of Im2Im-Deep and QUTCC across all five inverse problems. QUTCC consistently produces narrower uncertainty intervals. The black bolded line indicates the mean interval length.

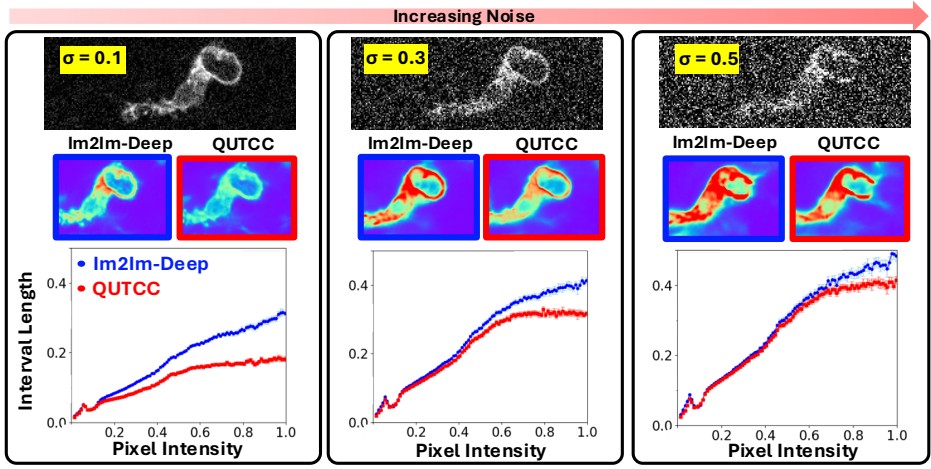

Figure 3: **QUTCC exhibits smaller uncertainty intervals in regions of high pixel intensity:** In a Gaussian denoising setting, we analyze how uncertainty interval lengths vary with pixel intensity across increasing noise levels ($\sigma = 0.1, 0.3, 0.5$). Under low-noise conditions, QUTCC exhibits substantially narrower uncertainty intervals compared to the baseline. As noise increases to $\sigma = 0.5$, this advantage becomes less pronounced overall. However, QUTCC continues to produce shorter uncertainty intervals for high-intensity pixels (intensity > 0.8), even at higher noise levels. All confidence intervals were estimated over a set of 10 samples.

## 4.2 UNCERTAINTY VISUALIZATIONS

Next, we visualize the predicted pixel-wise uncertainty of Im2Im-Deep and QUTCC for an undersampled MRI and a real noise image (Fig. 4). We compare these against the true error, which we obtain by taking the difference between the model prediction and the ground truth. Both Im2Im-Deep and QUTCC predict regions of high uncertainty that align with areas of high reconstruction error. However, QUTCC produces a more informative uncertainty map by selectively highlighting only the most uncertain regions, whereas Im2Im-Deep tends to assign broad, uniform areas of elevated uncertainty. In the MRI task shown in Fig. 4, arrows in the model prediction, uncertainty, and error maps indicate a hallucinated structure that appears in both the Im2Im-Deep and QUTCC predictions. QUTCC is able to accurately localize this hallucination with its uncertainty prediction, but Im2Im-Deep exhibits broader, less specific uncertainty across the surrounding structure.

## 4.3 ESTIMATING THE CONDITIONAL DISTRIBUTION

Finally, we demonstrate QUTCC's ability to construct a conformalized, pixel-wise conditional PDF in Fig. 5. To provide statistical coverage guarantees, we calibrate the model across a range of miscoverage levels $\alpha$ (see Suppl. Fig. 10 for visualization). By systematically varying $\alpha$ (e.g., from

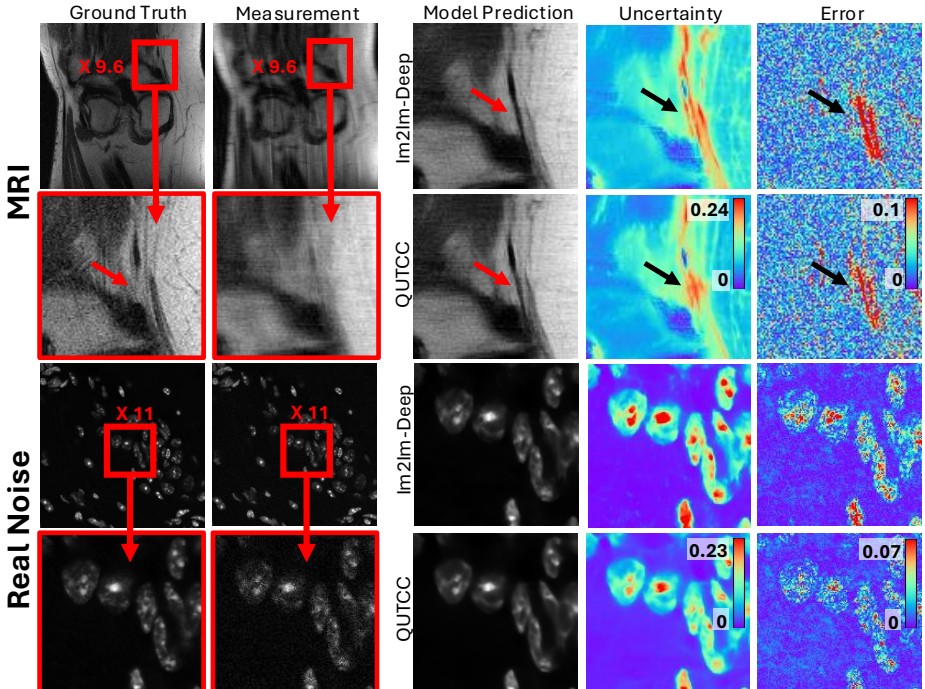

Figure 4: **Hallucination Visualization:** We show pixel-wise QUTCC and Im2Im-Deep uncertainty quantification for MRI and Real Noise tasks. In both imaging scenarios, both models highlight regions of high uncertainty that correspond to regions of high error. In the MRI task, the arrows point to a hallucination that appears in the Im2Im-Deep and QUTCC model predictions that is not present in the ground truth. QUTCC produces tighter uncertainty intervals that can better pinpoint uncertainty and hallucinations compared to Im2Im-Deep, which highlights a larger region.

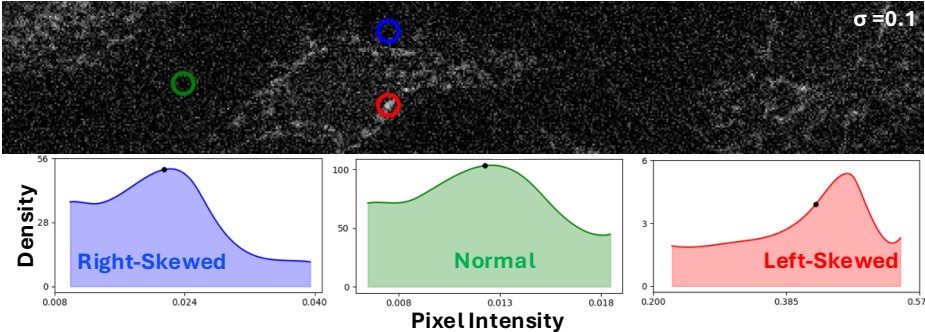

Figure 5: **QUTCC predicts diverse pixel-wise distributions:** By querying multiple quantiles and conformalizing the result, we can predict pixel-wise conditional distributions with statistical coverage guarantees. Here, we show several predicted conditional distributions for regions within an image given a measurement with Gaussian noise. At the blue, green, and red pixels, the model predicts a right-skewed distribution, a Gaussian-like distribution, and left-skewed, respectively.

0.1 to 0.9) and recording the corresponding quantile bounds, we obtain a collection of confidence intervals that, when aggregated, approximate the full cumulative distribution function. Differentiating this cumulative distribution function yields a conformalized pixel-wise PDF with formal coverage guarantees at each risk level. In Figure 5, we illustrate QUTCC 's ability to model diverse pixel-wise predictive distributions. The blue, green, and red patches show noticeably right, symmetric, and left-skewed distributions, respectively.

## 5   LIMITATIONS AND CONCLUSION

We propose QUTCC, a new uncertainty quantification method for imaging inverse problems that can achieve tighter uncertainty estimates than previous methods while maintaining the same statistical coverage. QUTCC accomplishes this by training a U-Net with a quantile embedding simultaneously on $q \in (0, 1)$ quantiles and then dynamically adjusting its quantile bound predictions during calibration until the desired risk is satisfied. We validated QUTCC on five imaging inverse problems - undersampled MRI, QPI, denoising under Gaussian, Poisson, and real noise, comparing its performance against prior conformal methods for image regression tasks. Our method exhibited tighter uncertainty intervals, on average, while still pinpointing model hallucinations and regions of high error. This can be attributed to our model learning and applying a nonlinear and asymmetrical scaling to its pixel-wise uncertainty predictions. Furthermore, QUTCC can estimate a conformalized conditional PDF, which previous conformal uncertainty quantification methods for image-to-image regression tasks could not do. While quantifying model uncertainty remains a significant open challenge in the field of deep learning, we believe that QUTCC offers a simple, yet robust method of uncertainty quantification for imaging inverse problems and image-to-image regression tasks. Some limitations are that QUTCC has a need for paired data for both the training and conformal calibration step. Additionally, we do not consider motion or 3D effects, which can be present in real samples. Interesting future work includes considering the effects of sample movement and distribution shifts, as well as uncertainty across multiple measurements instead of considering uncertainty for a single-frame independently. Additionally, exploring the integration of quantile conditioning into multi-hypothesis prediction architectures may enable the model to capture multiple plausible outcomes while leveraging the advantages provided by the learned quantiles.

**Ethical Considerations:** Using machine learning methods for scientific and medical applications has inherent risk - producing realistic artifacts that are not truly present in the image can be catastrophic for discovery and medical diagnostics. As research in uncertainty quantification matures, we hope that some of these risks can be mitigated to enable more trustworthy imaging. We acknowledge and fully comply with the ICLR Code of Ethics.

**Reproducibility:** We describe the different imaging inverse tasks in more detail in A.1. In this section, we also include the model epochs that we used to evaluate in this paper. Additional explanation of QUTCC's model architecture can be found in A.2, where we describe the quantile embedding in more detail. Code to train our model and reproduce the experiments shown here can be found at this anonymous code repo: **https://anonymous.4open.science/r/QUTCC-246F**.

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

# A   APPENDIX

These supplementary materials contain additional information to improve the reproducibility of the paper, as well as several additional analyses and experimental results. Code for training and analysis of QUTCC can be found at the following anonymous repo: **https://anonymous.4open.science/r/QUTCC-246F**. Section A.1 provides more details about the different inverse problems used throughout the paper and their associated datasets. Section A.2 details the network architecture used for QUTCC. Section A.3 provides additional comparisons against the original formulation of Im2Im-UQ without our network modifications and analysis on quantile crossing. Section A.4.1 includes analysis on the size-stratified risk, more visualizations of our uncertainty predictions across the different inverse problems, and a visualization of how QUTCC produces narrower uncertainty intervals through asymmetry. Finally, Sec. A.5 provides several visualizations of QUTCC's conditional distribution predictions, including an example showing how we conformalize the quantiles and the PDF prediction as a function of noise.

## A.1   EXPERIMENT DETAILS

In this section, we provide additional experimental details about our training process and datasets used for our five different tasks: denoising (real, Gaussian, Poisson), MRI, and QPI. Im2Im-Deep and QUTCC were each trained for 50 epochs on their respective datasets using a single NVIDIA RTX PRO 6000. Following training, we conducted a model selection sweep to identify the epoch that yielded the narrowest uncertainty intervals while satisfying the target risk level ($\alpha = 0.1$).

For all models, an initial learning rate of 1e-4 and weight decay of 0 was used. Batch size was adjusted depending on the task, with 4 used for denoising tasks, 12 for MRI, and 72 for QPI for U-Net backbones and 32, 512, and 16 (respectively) for ResNet-18 backbones. Images were normalized before training.

### A.1.1   REAL NOISE TASK

For the real noise task, we used the Fluorescence Microscopy Denoising (FMD) dataset (Zhang et al., 2019), which contains experimentally acquired fluorescence microscopy images spanning 12 wide-field, confocal, and two-photon modalities. The model was trained on 10,000 images, with 500 confocal mouse images used for calibration and an additional 500 for validation. The epochs used for Im2Im-Deep and QUTCC are 5 and 10, respectively.

### A.1.2   GAUSSIAN AND POISSON NOISE TASK

For both the Gaussian and Poisson tasks, we synthetically introduced varying levels of noise to the FMD ground truth images. The dataset was split into 180 ground truth images for training, 40 for calibration, and 20 for validation. The pseudocode for generating gaussian and poisson noise are shown below in Algorithm 2 and Algorithm 3.

---

**Algorithm 2** Add Gaussian Noise

1: **Input:** Clean image $\mathbf{x}$, max noise level $\sigma_{\max}$
2: **Output:** Noisy image $\mathbf{y}$
3: Sample noise std: $\sigma \sim \mathcal{U}(0, \sigma_{\max})$
4: Sample Gaussian noise: $\boldsymbol{\eta} \sim \mathcal{N}(0, \sigma^2)$
5: Add noise: $\mathbf{y} \leftarrow \mathbf{x} + \boldsymbol{\eta}$
6: Clamp: $\mathbf{y} \leftarrow \mathrm{Clamp}(\mathbf{y}, 0, 1)$
7: **return** $\mathbf{y}$

---

**Algorithm 3** Add Poisson Noise

1: **Input:** Clean image $\mathbf{x}$, min/max noise levels $\sigma_{\min}, \sigma_{\max}$
2: **Output:** Noisy image $\mathbf{y}$
3: Sample scale: $\lambda \sim \mathcal{U}(\sigma_{\min}, \sigma_{\max})$
4: Scale image: $\mathbf{x}_{\mathrm{scaled}} \leftarrow \lambda \cdot \mathbf{x}$
5: Sample noise: $\boldsymbol{\eta} \sim \mathrm{Poisson}(\mathbf{x}_{\mathrm{scaled}})$
6: Clamp: $\mathbf{y} \leftarrow \mathrm{Clamp}(\boldsymbol{\eta}, 0, 1)$
7: **return** $\mathbf{y}$

---

For Gaussian noise, we set $\sigma_{\max} = 0.5$. For Poisson noise, the noise level range $(\lambda_{\min}, \lambda_{\max})$ was set to $(50, 100)$. In each iteration, with a batch size of 16, we generated 25 random noise levels uniformly sampled between the specified minimum and maximum values. The number of training

epochs for Im2Im-Deep and QUTCC with Gaussian noise were 15 and 20, respectively. For Poisson noise, Im2Im-Deep was trained for 10 epochs, while QUTCC was trained for 35 epochs.

### A.1.3 Magnetic Resonance Imaging (MRI) Task

Data used for the MRI task was obtained from the NYU fastMRI initiative database (fastmri.med.nyu.edu) (Knoll et al., 2020; Zbontar et al., 2018). The primary goal of fastMRI is to test whether machine learning can aid in the reconstruction of medical MRI images. To train our models, we split the fastMRI dataset into 700 volumes for training, 200 volumes for calibration, and 200 for validation. It is important to note that a single volume contains multiple MRI images. The epochs used for Im2Im-Deep and QUTCC are 25 and 30, respectively.

To simulate the forward model in MRI, we start with a fully-sampled 3D volume composed of multiple 2D image slices. Each 2D image slice is transformed into its frequency domain representation using the 2D Fourier Transform, producing its k-space data. To simulate undersampled acquisition, we apply a $4\times$ undersampling mask to the k-space. The resulting undersampled k-space is then transformed back into the image domain using the inverse Fourier Transform, yielding an aliased or artifact-corrupted image that serves as the input for the models.

### A.1.4 Quantitative Phase Imaging (QPI) Task

Data used for the QPI task were obtained from the Berkeley Single Cell Computational Microscopy (BSCCM) dataset (Pinkard et al., 2024). The BSCCM dataset contains image samples of individual white blood cells that have been captured with several illumination patterns on an LED array microscope. For this task, we used 289,059 images for training, 82,588 images for calibration, and 41,294 images for validation. The number of training epochs for Im2Im-Deep and QUTCC were 15 and 20, respectively.

In QPI, the goal is to image the structure of transparent cells by recovering the phase delay of the light that passes through the cells. Several intensity-only images are used to computationally estimate the phase of the light, since this cannot be measured directly. The input measurement $\mathbf{y}$ is the concatenation of two cell intensity images acquired at different illumination angles. The corresponding ground truth is the quantitative phase image recovered from four or more illumination angles. The model is trained to map these two intensity images into a phase image.

### A.2 Model Architecture

Our method, QUTCC, is based on a U-Net backbone augmented with self-attention mechanisms, where quantile embeddings are propagated through the self-attention layers to guide the network's quantile predictions. In our design, the target quantile level is embedded as a continuous scalar, analogous to the time-step embeddings in diffusion models. A core part of this architecture is the integration of self-attention layers within the U-Net, implemented as `AttentionBlock` modules. These blocks allow the model to capture global dependencies across spatial dimensions, enabling each pixel or feature location to attend to all other locations within its feature map. Specifically, attention layers are incorporated at various downsampling resolutions within both the encoder and decoder paths, as well as at the bottleneck of the U-Net.

The ability to condition the model's output on a given quantile is achieved through the quantile embedding mechanism. A quantile value between (0, 1) is chosen for each image sample at each iteration and then transformed into a high-dimensional vector representation using a positional encoding scheme, employing sinusoidal functions to create generalizable embeddings. This initial embedding is then processed by a small multi-layer perceptron, which is then used throughout to condition the network. By introducing this conditioning, the model learns to generate outputs that are responsive to the entire range of quantile levels. This mechanism works in tandem with the self-attention layers, which are specified using the `attention_resolutions` parameter.

In our experiments, QUTCC was trained using `attention_resolutions` configured as $[16, 8, 4, 2, 1]$. This means for 512 x 512 images, using the specified configuration results in attention layers within the encoder path (at resolutions 512x512, 256x256, 128x128, 64x64, and 32x32), one central attention layer in the middle block (at 8x8 resolution), and additional attention layers distributed across the decoder path (at resolutions 32x32, 64x64, 128x128, 256x256,

and 512x512). The `attention_resolutions` parameter dictates the spatial scales at which these attention mechanisms are introduced. Further model configurations can be found in `models/model_config.yaml`.

## A.3 MODEL ANALYSIS

In Fig. 4, we restricted our comparisons to the Im2Im-Deep and QUTCC models. The original Im2Im-UQ model was excluded due to its comparatively shallow architecture, resulting in decreased performance. However, for the subsequent analysis, we reintroduce Im2Im-UQ for completeness. In this section, we assess the model's mean predictive performance and the quantile crossing occurrences.

### A.3.1 PREDICTION PERFORMANCE

Does QUTCC produce tighter intervals because it is simply a better image prediction network? To investigate this, we compare the predictive performance of Im2Im-UQ, Im2Im-Deep, Im2Im-Deep-Median, and QUTCC. Im2Im-Deep-Median is a variant of Im2Im-Deep that predicts the median rather than the mean. Since QUTCC's estimates are centered on the median, we train this median-predicting version of Im2Im-Deep to ensure a fair comparison. In Table 2, we compare the performance of Im2Im-Deep and QUTCC using standard image reconstruction metrics: MSE, SSIM, PSNR, and LPIPS. For QUTCC, the mean prediction was obtained by setting the quantile level to $q = 0.5$. The results indicate that all models achieve nearly identical performance in terms of MSE, with only minor differences observed in SSIM, PSNR, and LPIPS. These variations are not substantial enough to suggest that QUTCC provides a significantly better mean prediction. These findings suggest that QUTCC 's improved uncertainty quantification predictions are not attributed to better mean prediction performance. Rather, its ability to more effectively characterize uncertainty appears to come from the explicit learning of quantiles during training.

| Metric | Model | MRI | QPI | Gaussian | Poisson | Real Noise |
|---|---|---|---|---|---|---|
| MSE (↓) | Im2Im-UQ | $0.003 \pm 0.002$ | $0.0006 \pm 0.0005$ | $0.0007 \pm 0.0006$ | $0.0003 \pm 0.0003$ | $0.0030 \pm 0.0004$ |
| | Im2Im-Deep | $0.001 \pm 0.002$ | $0.0004 \pm 0.0003$ | $0.0006 \pm 0.0006$ | $0.0003 \pm 0.0002$ | $0.0004 \pm 0.0002$ |
| | Im2Im-Deep-Median | $0.001 \pm 0.002$ | $0.0003 \pm 0.0002$ | $0.0006 \pm 0.0002$ | $0.0003 \pm 0.0002$ | $0.0006 \pm 0.0004$ |
| | QUTCC | $0.001 \pm 0.002$ | $0.0004 \pm 0.0003$ | $0.0006 \pm 0.0005$ | $0.0003 \pm 0.0003$ | $0.0002 \pm 0.0001$ |
| SSIM (↑) | Im2Im-UQ | $0.668 \pm 0.127$ | $0.949 \pm 0.017$ | $0.852 \pm 0.102$ | $0.931 \pm 0.038$ | $0.803 \pm 0.016$ |
| | Im2Im-Deep | $0.707 \pm 0.139$ | $0.961 \pm 0.010$ | $0.856 \pm 0.107$ | $0.937 \pm 0.035$ | $0.959 \pm 0.006$ |
| | Im2Im-Deep-Median | $0.708 \pm 0.139$ | $0.961 \pm 0.010$ | $0.865 \pm 0.101$ | $0.932 \pm 0.039$ | $0.952 \pm 0.013$ |
| | QUTCC | $0.708 \pm 0.139$ | $0.959 \pm 0.010$ | $0.865 \pm 0.102$ | $0.941 \pm 0.036$ | $0.957 \pm 0.008$ |
| PSNR (↑) | Im2Im-UQ | $26.867 \pm 2.923$ | $33.135 \pm 2.543$ | $32.739 \pm 3.535$ | $36.163 \pm 3.059$ | $25.833 \pm 0.734$ |
| | Im2Im-Deep | $29.711 \pm 3.138$ | $34.565 \pm 2.393$ | $33.557 \pm 4.018$ | $37.062 \pm 2.968$ | $34.038 \pm 1.837$ |
| | Im2Im-Deep-Median | $29.744 \pm 3.130$ | $35.191 \pm 2.398$ | $33.708 \pm 4.244$ | $36.576 \pm 2.824$ | $33.388 \pm 3.646$ |
| | QUTCC | $29.833 \pm 3.156$ | $34.948 \pm 2.436$ | $33.660 \pm 4.143$ | $37.498 \pm 3.797$ | $37.350 \pm 1.936$ |
| LPIPS (↓) | Im2Im-UQ | $0.343 \pm 0.033$ | $0.153 \pm 0.025$ | $0.420 \pm 0.092$ | $0.294 \pm 0.071$ | $0.360 \pm 0.033$ |
| | Im2Im-Deep | $0.324 \pm 0.043$ | $0.125 \pm 0.015$ | $0.414 \pm 0.103$ | $0.299 \pm 0.071$ | $0.297 \pm 0.029$ |
| | Im2Im-Deep-Median | $0.322 \pm 0.040$ | $0.121 \pm 0.015$ | $0.405 \pm 0.097$ | $0.304 \pm 0.071$ | $0.324 \pm 0.027$ |
| | QUTCC | $0.323 \pm 0.040$ | $0.121 \pm 0.015$ | $0.408 \pm 0.102$ | $0.284 \pm 0.072$ | $0.312 \pm 0.026$ |

Table 2: **Image reconstruction performance of Im2Im-UQ, Im2Im-Deep, Im2Im-Deep-Median, and QUTCC**: For each metric, the arrow indicates the direction of better performance.

### A.3.2 QUANTILE CROSSING PERFORMANCE

In section 3.4 we describe the conformal calibration step, which is dependent on the quantile function being monotonic. To ensure the validity of the predicted quantiles, specifically to avoid the issue of quantile crossing, we quantified the number of quantile crossing occurrences between $q = [0.1, 0.2, 0.3, ..., 0.9]$ in QUTCC (Tbl 3a). Quantile crossing can undermine the interpretability of our uncertainty estimates, as it contradicts the notion that quantile functions should be non-decreasing/non-overlapping. The results indicate that across all imaging tasks, the ratio of quantile crossing occurrences is minimal.

We additional compare the quantile crossing occurrences between Im2Im-UQ and QUTCC, utilizing each model's respective bounds, which can be visualized in Tbl 3b, observing comparable performance between both models.

| Task | Crossed Pix. | Total Pix. | Cross. Ratio |
|------|-------------|------------|--------------|
| MRI | 2.20e1 | 1.64e9 | 1.34e−8 |
| QPI | 5.00 | 2.62e8 | 1.91e−8 |
| Gaussian | 1.10e6 | 3.36e9 | 3.29e−4 |
| Poisson | 3.49e3 | 3.36e9 | 1.04e−6 |
| Real Noise | 3.38e4 | 1.05e9 | 3.23e−5 |

| Task | Im2Im | QUTCC | Total Pix. |
|------|-------|-------|------------|
| MRI | 0 | 0 | 2.05e8 |
| QPI | 0 | 0 | 3.28e7 |
| Gaussian | 3.91e3 | 3.30e3 | 4.19e8 |
| Poisson | 7.00 | 2.47e2 | 4.19e8 |
| Real Noise | 0 | 0 | 1.31e8 |

(a) QUTCC Quantile crossing occurrences for quantiles [0.1, 0.2, 0.3, ..., 0.9]

(b) Quantile crossing for upper and lower bounds of Im2Im vs QUTCC

Table 3: **Quantile crossing analysis**

### A.3.3 QUANTILE EMBEDDING ABLATION

To validate our design choice of conditioning every residual block of our model with quantile embeddings, we evaluate the impact of restricting these embeddings to specific network components. We train six variants of our model on the Gaussian denoising task, where each variant keep the same architecture and optimization but restricts conditioning to a subset of the residual blocks. We test four competing hypotheses regarding where quantile information is more critical: (i) that the model only requires embedded feature extraction in the encoder; (ii) that the model only requires embeddings in the decoder; (iii) that the model only requires semantic conditioning at the bottleneck; and (iv) that the model only requires conditioning at the highest-resolution layers (input and/or output).

We report the mean interval length and risk coverage of each calibrated model, targeting $\alpha = 0.1$, in Table 4. The baseline configuration achieves the tightest intervals while maintaining valid risk coverage. Notably, restricting conditioning to the early layers or exclusively to the high-resolution layers results in poor calibration, suggesting that the model is unable to propagate quantile-specific information to the final output. Conversely, the model calibrates to a valid $\alpha$ but produces overly conservative intervals when conditioning on only the bottleneck or decoder. This suggests that the network learns more efficient features and can use them more effectively when quantile information is injected at multiple scales throughout the architecture.

| Model | Interval Length | Total Risk |
|-------|----------------|------------|
| **Embedding in all blocks (ours)** | **0.0592** | **0.0908** |
| Embedding in first block only | 0.0619 | 0.1243 |
| Embedding in encoder blocks only | 0.0645 | 0.1199 |
| Embedding in middle block only | 0.0695 | 0.0829 |
| Embedding in decoder blocks only | 0.0679 | 0.0895 |
| Embedding in final block only | 0.0648 | 0.0987 |
| Embedding in first and final blocks | 0.0676 | 0.1012 |

Table 4: **Quantile Embedding Ablation**: We evaluate the impact of quantile conditioning at different stages of our U-Net architecture on the Gaussian denoising task. Our best-performing baseline model applies the embedding to every residual block. We report the mean interval length and calibrated risk targeting $\alpha = 0.1$.

### A.4 ADDITIONAL UNCERTAINTY INTERVAL EVALUATIONS

We stratify the uncertainty interval size of all the inverse tasks by pixel intensity (Fig. 6). We demonstrate that while Im2Im, its variants, and QUTCC produce similarly sized uncertainty intervals in low-signal (background) regions, QUTCC learns to produce noticeably smaller intervals in regions of high pixel intensity. We also introduce Im2Im-Asymm., an Im2Im baseline with asymmetric calibration in which the upper and lower bounds use different lambda scales. QUTCC generally outperforms Im2Im-Asymm. as well, indicating that its performance gains arise not merely from asymmetric calibration but from learning the full quantile distribution. In Gaussian, Poisson, and

real noise there is a noticeably larger decrease in QUTCC interval size as pixel intensity increases. This trend is less consistent for the MRI and QPI tasks, but QUTCC still generally produces smaller interval lengths across most stratified intensity bins.

### A.4.1 Pixel Intensity Stratified Uncertainty Interval Lengths

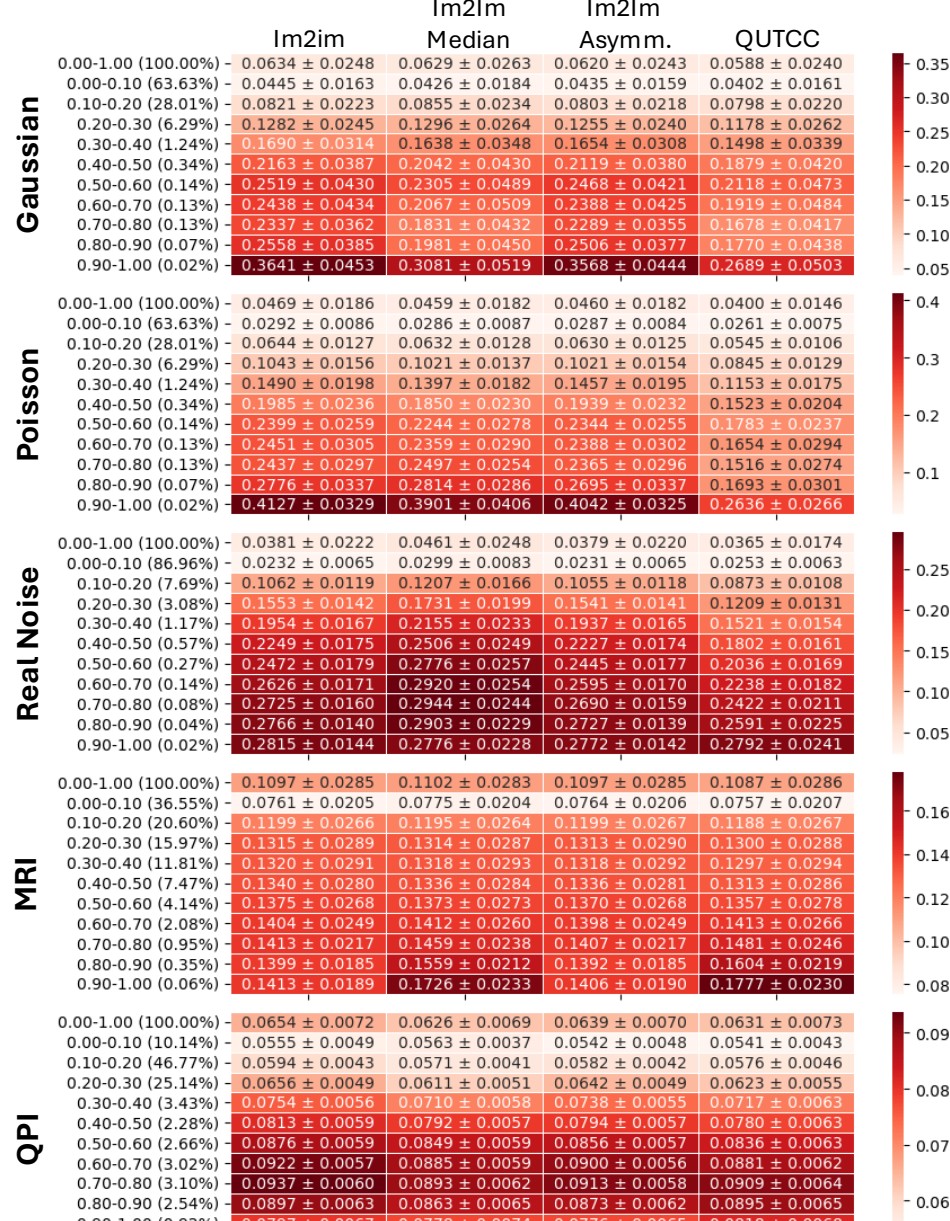

Figure 6: **Pixel Intensity Stratified Uncertainty Interval Sizes**: We compare the uncertainty interval lengths of Im2Im, its variants, and QUTCC across all five inverse tasks. For each dataset, the first row reports the overall average uncertainty, and the subsequent rows present interval lengths stratified by pixel intensity from 0 to 1 in increments of 0.1.

### A.4.2 Size-Stratified Risk

We observed the size-stratified risk of all inverse tasks between Im2Im-Deep and QUTCC (Fig. 7). To calculate size-stratified risk, the prediction intervals are first binned into different sizes, ranging from

smallest to largest. Then the risk is calculated across all the bins to ensure that the model's uncertainty estimates are well-calibrated across different levels of confidence. While both Im2Im-Deep and QUTCC have bins that exceed the $\alpha$, generally, most bins fall under the chosen risk.

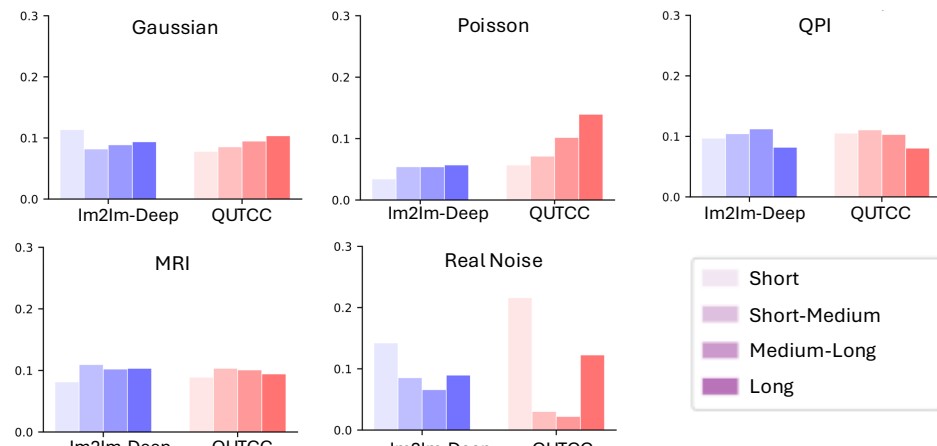

Figure 7: **Size-Stratified Risk of Im2Im-Deep vs. QUTCC**: We evaluate the size-stratified risk of Im2Im-Deep and QUTCC across all tasks. Overall, neither model exhibits a strong relationship between interval width and empirical risk, suggesting limited sensitivity to interval size. However, in the Gaussian and Poisson settings, both models display a mild trend toward improved calibration, or lower risk, for narrower prediction intervals.

### A.4.3 ADDITIONAL VISUALIZATIONS

We also provide visualizations of the remaining imaging tasks not included in the main results (Fig. 8). For all sample tasks, both QUTCC and Im2Im-Deep effectively highlight regions with high reconstruction error. However, QUTCC demonstrates a more focused identification of areas with high uncertainty that align closely with the true error, indicating its greater precision in uncertainty estimation. While Im2Im-Deep is capable of identifying regions of error, it tends to assign elevated uncertainty across larger portions of the sample, making it challenging for downstream tasks to prioritize regions based on uncertainty interval sizes. This trend is consistent across all five imaging inverse problems.

### A.4.4 BOUND ASYMMETRY

How does QUTCC achieve tighter confidence intervals while maintaining the same level of risk? As shown in Fig. 9, QUTCC produces asymmetric predictive intervals— its upper and lower bounds are adjusted independently based on localized uncertainty. In contrast, Im2Im-UQ applies a single global scaling factor $\lambda$ uniformly to both bounds, which can be suboptimal in cases where only one side of the interval requires adjustment. This limitation often leads to unnecessarily widened intervals. In the red boxed region of Fig. 9, both QUTCC and Im2Im-UQ share a similar lower bound, yet QUTCC predicts a significantly tighter upper bound. Similarly, in the green boxed region, both methods align on the upper bound, but QUTCC yields a tighter lower bound. These examples highlight QUTCC's ability to adaptively adjust its interval predictions, leading to more precise interval estimates.

### A.5 ADDITIONAL PDF RESULTS

In this section, we present additional results highlighting QUTCC's ability to estimate a conditional probability density function. In Fig. 10a, we show several of the predicted PDFs for image denoising with Gaussian noise with $\sigma = 0.4$. Detailed views of the corresponding pixel-wise PDFs are presented in Fig. 10b for the low-intensity pixel and Fig. 10c for the high-intensity pixel. Each graph displays two PDFs: the blue PDF represents the quantile predictions from the uncalibrated model, while the green PDF represents the conformally calibrated quantiles that provide finite-sample statistical

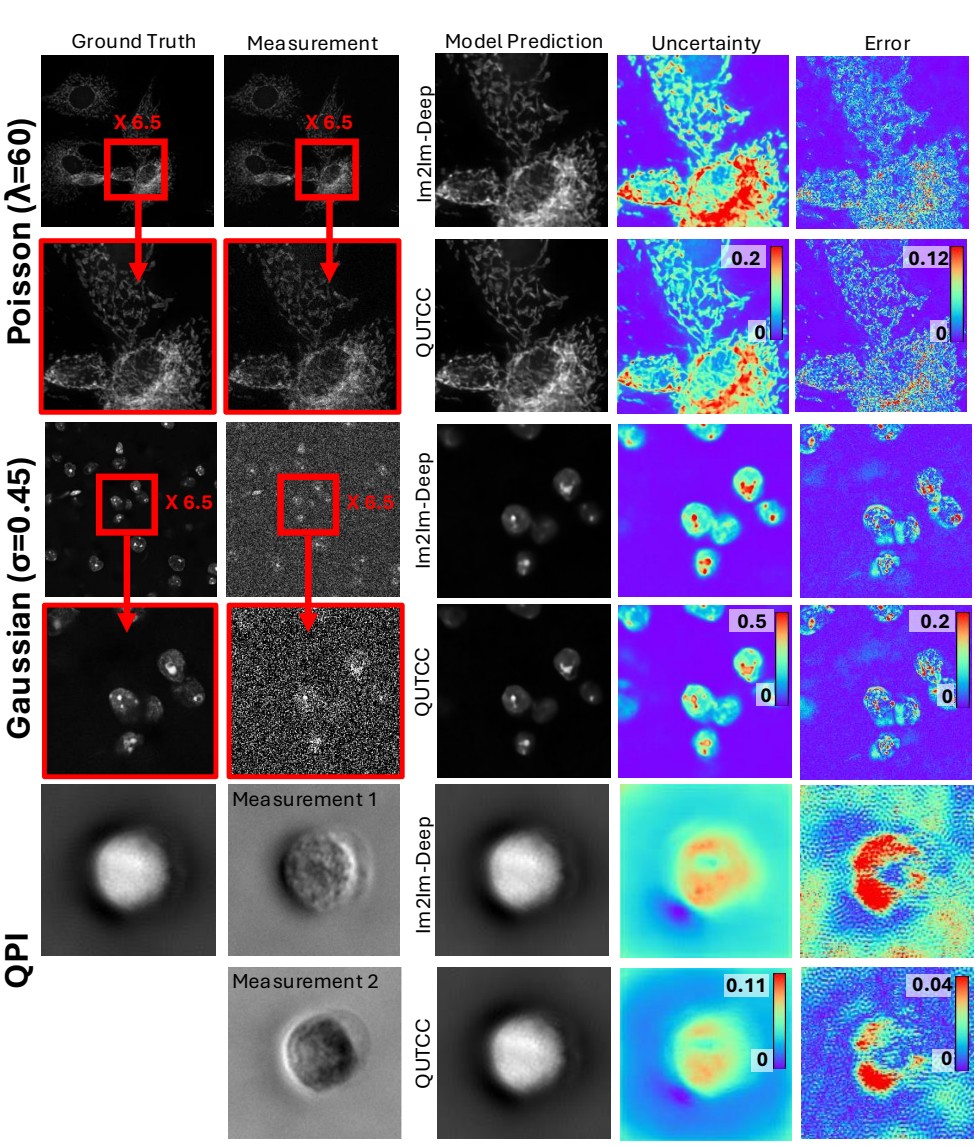

Figure 8: **Additional Uncertainty Visualizations:** We visualize both the full and zoomed-in regions of image reconstructions for QPI and denoising with Poisson, Gaussian and Real Noise. Consistent with observations presented in the results section, QUTCC produces more precise uncertainty estimates that closely align with localized regions of high reconstruction error. In contrast, Im2Im-Deep tends to highlight broader regions of uncertainty and lacks specificity, making it hard to distinguish areas of importance. This highlights QUTCC 's ability predict more informative uncertainty maps.

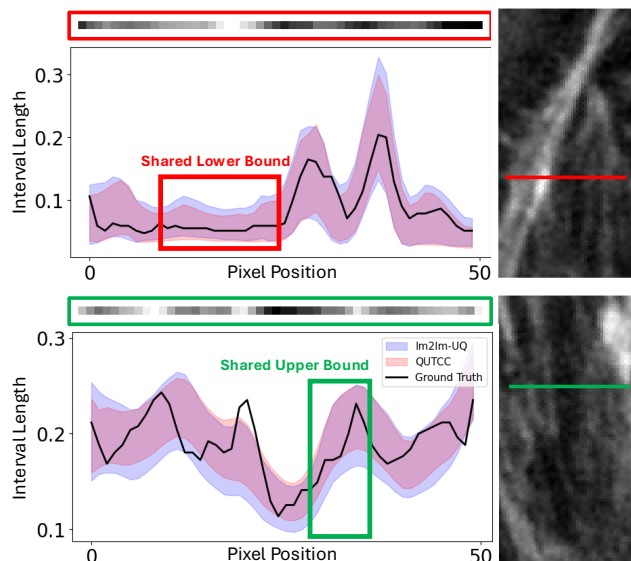

Figure 9: **QUTCC produces narrower intervals through asymmetric bounds**: We analyze the pixel-wise uncertainty bounds predicted by Im2Im-UQ and QUTCC and observe that QUTCC exhibits asymmetric behavior in its interval estimates. In Im2Im-UQ, both the upper and lower bounds are uniformly scaled by a global factor $\lambda$ to satisfy coverage constraints, which limits flexibility in adapting to signal-specific uncertainty. In contrast, QUTCC learns to predict quantiles directly, enabling it to independently modulate upper and lower bounds based on signal characteristics. This results in more adaptive and efficient uncertainty intervals. For instance, in the red boxed region, QUTCC matches Im2Im-UQ's lower bound but produces a significantly tighter upper bound. Conversely, in the green boxed region, both models share an upper bound, yet QUTCC yields a tighter lower bound. Samples shown are Gaussian images with $\sigma = 0.1$.

coverage guarantees. The conformal calibration procedure adjusts the quantile levels to ensure valid coverage properties. For instance, while the 25th and 75th percentiles theoretically provide $50\%$ coverage, conformal calibration determines that the 20.7th and 69.2nd percentiles are required to achieve statistically guaranteed $50\%$ coverage for this specific dataset and model.

Additionally, Fig. 11 illustrates the evolution of the pixel-wise PDF for a fixed pixel coordinate under varying Gaussian noise levels $\sigma \in \{0.1, 0.3, 0.5\}$. As the noise standard deviation increases, the PDF widens. This widening corresponds to increased epistemic uncertainty in the model's predictions, as higher noise levels reduce the information content available for accurate pixel intensity estimation. This then directly widens prediction intervals to maintain the desired coverage guarantees.

### A.5.1 PIXELWISE PDF CHANGES AS NOISE DISTRIBUTION CHANGES

We examine how the pixelwise PDFs change as the underlying noise distribution varies. In Fig. 12, we show image predictions for measurements with differing levels of Gaussian and Poisson noise. Images recovered from measurements with Poisson noise have a distinct skewed shape that we associate with a Poisson distribution, whereas the Gaussian PDFs are more symmetric at higher intensity values before clipping is observed at the lower intensity values. We can see that the predicted PDFs broaden as the noise standard deviation increases. Joint reconstruction and PDF estimation could be useful to assess the underlying noise distribution present in an image, and estimating the PDF has been useful for a number of tasks in microscopy Krull et al. (2020).

### A.5.2 IMPACT OF QUANTILE NUMBER ON ESTIMATED PDF

We investigate how the number of quantiles influences the predicted PDF. The PDF is estimated from the numerical derivative of the quantile function. This estimate depends on the spacing between adjacent quantiles. Figure 13 illustrates this effect by showing PDFs reconstructed using progressively more quantile samples. The overall shape stays relatively similar as the number of quantile queries increases.

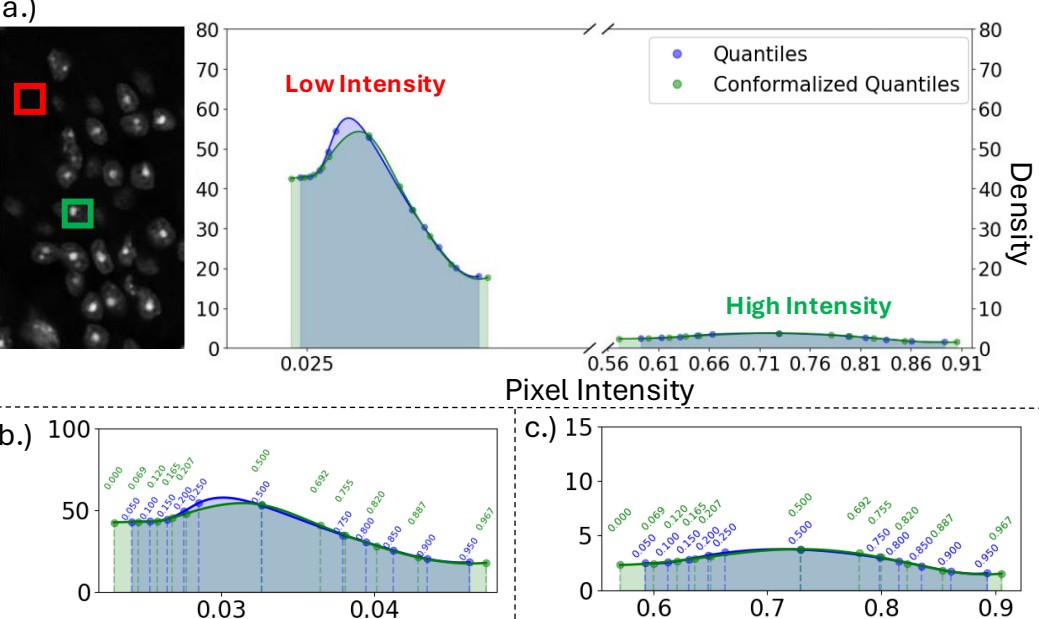

Figure 10: **QUTCC predicts different pixel-wise PDFs based on different siganl intensity a)** Comparison of pixel-wise PDFs for representative low-intensity and high-intensity pixels in a Gaussian measurement ($\sigma = 0.4$). **b)** Detailed view of the low-intensity pixel PDF, exhibiting a narrow, high-density distribution concentrated around few intensity values, indicating low predictive uncertainty. **c)** Detailed view of the high-intensity pixel PDF, showing a broader, lower-density distribution with increased spread, reflecting higher predictive uncertainty in bright image regions. The blue lines show the uncalibrated model, while the green lines show the conformalized quantiles after calibration.

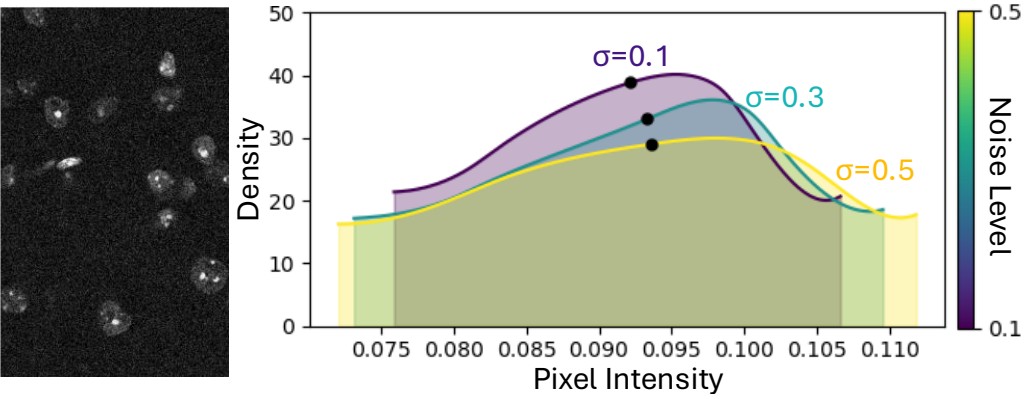

Figure 11: **PDF broadens as noise increases:** We observe the PDF of a single pixel under varying noise levels. At $\sigma = 0.1$, the noise is low, and the PDF is compact. As the noise increases to $\sigma = 0.3$ and $\sigma = 0.5$, the PDF gradually broadens, while the mean prediction value remains relatively unchanged. This broadening occurs due to the increased uncertainty introduced by higher noise levels. QUTCC successfully predicts this increased uncertainty as the noise increases.

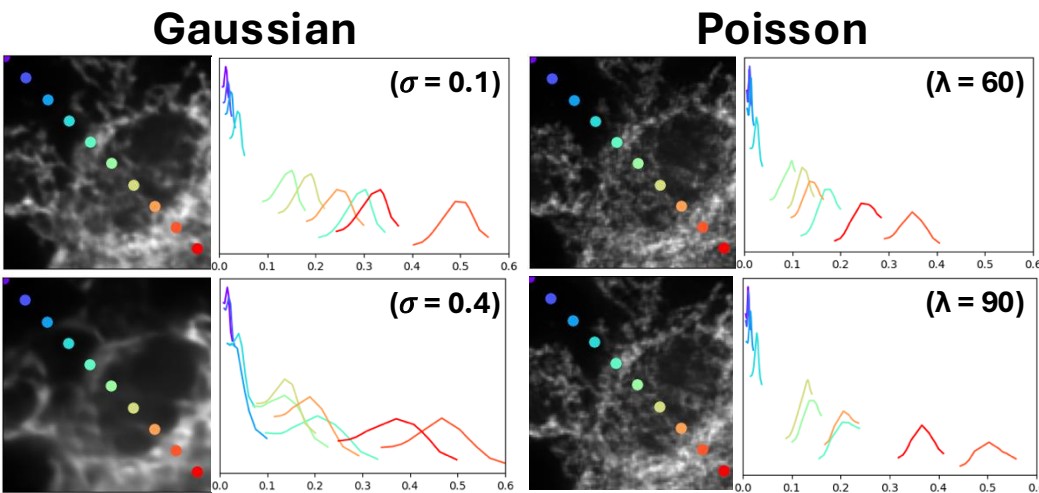

Figure 12: **Predicting Pixelwise PDFs:** We compare the predicted probability density functions (PDFs) of 10 representative pixels recovered from a measurement with different levels of Gaussian and Poisson noise.

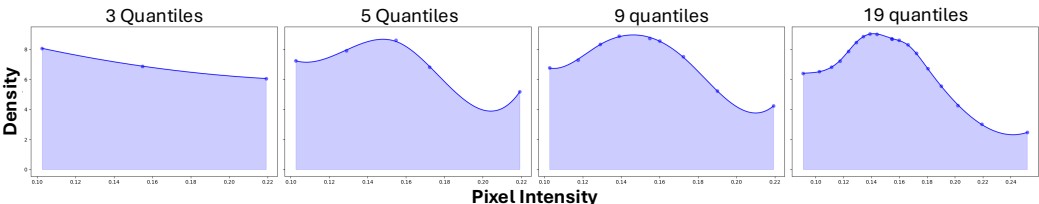

Figure 13: **Pixelwise PDF as a function of queried quantiles**: We demonstrate the changes in PDF shape as the number of queried quantiles increase.

### A.6 LLM USAGE

LLMs were used to edit and correct grammar during writing, but were not involved in the research ideation process.

