# OpenReview forum: "QUTCC: Quantile Uncertainty Training and Conformal Calibration for Imaging Inverse Problems"
_ICLR.cc/2026/Conference — Submitted to ICLR 2026_

### Official Review · Reviewer_ak8f · 2025-10-25

**Soundness:** 2
**Presentation:** 2
**Contribution:** 3
**Rating:** 2
**Confidence:** 4

**Summary:**

The submission focuses on conformal risk control for imaging inverse problems. The submission proposes to train a quantile-conditional neural network to jointly estimate all quantiles $q \in [0,1]$ of the pixel-wise conditional distribution of the ground-truth image given a noisy observation. With this network, the submission introduces an algorithm to estimate the conformalized quantile $\hat{q}$ that guarantees coverage (in the conformal risk control sense) of the ground-truth image.

Experiments compare the proposed method with one existing baseline method.

**Strengths:**

* Uncertainty volume minimization in high-dimensional inverse problems is an important problem.
* Training a quantile-conditional network is an interesting idea.
* Estimating the full posterior density using quantile regression is novel in the context of imaging inverse problems.

**Weaknesses:**

* Presentation of theory is hand-wavy.
* Experimental results provide limited evidence in support of the proposed method.

I will expand on a few questions below and I am looking forward to discussing with the authors!

**Questions:**

**Conformal Risk Control vs. Risk-Controlling Prediction Sets**

In its current formulation, there appears to be a disconnect between the guarantee mentioned in the text, the one written in Eq. (2), and the one provided by the algorithm. If I understand correctly:

* $x,y$ are high-dimensional objects, let us say $\mathbb{R}^d$ (this should be specified clearly in the text)
* Lines 183-185 state that the prediction set should "contain at least $1 - \alpha$ of the ground-truth pixels with probability $1- \delta$". This is the notion of risk-controlling prediction set of [1] (Definition 1)

$$\mathbb{P}[ \mathbb{E}[\frac{1}{d} \sum_{j \in [d]} \mathbb{I}[x_j \notin C_{\lambda}(y)_j]] \leq \alpha]\geq 1 - \delta.$$

* However, Eq. (2) requires the **entire** input to be contained at once in the prediction set, which is a stronger guarantee. In practice, Eqs. (5) and (6) refer to the risk-controlling notion, not Eq. (2).

* Finally, the algorithm implements a conformal risk control procedure that does not provide a high-probability claim, but in expectation only (Algorithm 1 does not depend on $\delta$). That is, the guarantee provided by conformal risk control is:

$$\mathbb{E}[\frac{1}{d} \sum_{j \in [d]} \mathbb{I}[x_j \notin C_{\lambda}(y)_j]] \leq \alpha.$$

In fact, the submission uses the same procedure described in [2], Eq. (4). Note that to achieve the stronger notion of high-probability risk control, binary search would not be a valid algorithm, but other strategies, such as backtracking or fixed hypothesis testing might be used [3].

Could the authors clarify this apparent inconsistency?

**Multiple quantile estimation and quantile crossing**

The submission acknowledges that for the proposed method to be statistically valid, the risk must be monotonically non-increasing with the quantile level. This implies that no quantile crossing can happen, but the training procedure proposed in the submission does not prevent quantile crossing by design. In fact, Table 3 in the appendix shows that quantile crossing does happen (even if minimally).

Experimentally, this does not seem to affect the validity of the calibration procedure, but it renders the guarantees of the proposed method vacuous in the general sense.

Could the authors propose technical ways to address this robustly? For example, see [4], who also propose to learn multiple quantiles simultaneously in order to estimate conformalized histograms of the posterior distribution (related to the goal of Sec. 3.4.1, although different in methodology and application).

Could the authors include a formal statement of the guarantees provided by the proposed method? This will improve clarity of presentation, even if the result might follow from standard properties of conformal prediction.

**Definition of the loss function**

In its current formulation, the loss in Eq. (4) does not faithfully represent the sampling of the quantile levels from a uniform distribution.

Could the authors state the loss function as an expectation over a population, clearly stating the distribution quantile levels are sampled from at training time?

**Estimating the conditional distribution**

It might be helpful to distinguish between the true quantile function, the empirical quantile function, and the estimated quantile from the trained network. Note that in the general sense, it is not true that the quantile function is the inverse of the CDF, as the latter might not be strictly increasing.

I am not sure I understand what coverage guarantee the estimated PDF provides. Could the authors expand on this point and clearly state the guarantee(s)? Is this similar to the concept of conformal predictive distributions [5]?

As an aside, it might be worth mentioning existing Bayesian approaches that provide estimates of these distributions by design, with, for example, Gaussian processes.

**Experimental results**

Table 1 and Figure 2 provide weak evidence that the proposed method significantly provides shorter intervals compared to im2im, as all differences are well within one standard deviation of the results. The result in Figure 3 is more convincing. Could the authors provide the equivalent of Figure 2 stratified by pixel intensity? That might be a more compelling example of the advantages provided by the proposed method.

If the main advantage of the proposed method is to provide shorter interval lengths, it is important to compare with existing works that address this in imaging inverse problems [6,7].

I am not sure I follow the choice of training two different models, one for im2im and one for QUTCC. If I understand correctly, one could use im2im to calibrate the predictions of a network trained with multiple quantile regression as well. Could the authors expand on the choice of using different networks instead of keeping the network fixed and changing the calibration procedure only? That might provide a more straightforward comparison between the two methods.

---

**Minor comments**

* "Where accuracy is more important than perceptual quality" may be too broad in its current formulation. Image quality evaluation is a nuanced and complementary field of research, especially in medical applications. Uncertainty quantification is a well-motivated endeavor even without this comment.
* The emoji in the title seems unrelated to the contribution of the paper? I would consider removing unless it contributes to the submission.
* Typo in Eq. (8), $\geq 1-\alpha$?
* In its current formulation, does Algorithm 1 always terminate? Since the step size does not change, isn't there a chance the thresholds will start oscillating?
* The claim that no existing methods have addressed minimal interval length or conditional coverage in imaging inverse problems is too strong as there exist prior works that have proposed ways to tackle parts of these issues [6,8].
* The claim that prior works use linear scaling of the interval bounds might also be too broad, as several works have used additive parametrizations that do not scale intervals bounds multiplicatively, or different strategies such as masking.

---

**References**

[1] Angelopoulos et al. "Image-to-Image Regression with Distribution-Free Uncertainty
Quantification and Applications in Imaging" (2022).

[2] Angelopoulos et al. "Conformal Risk Control", (ICLR 2024).

[3] Angelopoulos et al. "Learn then Test: Calibrating Predictive Algorithms to Achieve Risk Control", (2022).

[4] Sesia and Romano "Conformal Prediction using Conditional Histograms", (2021).

[5] Vovk et al. "Conformal Predictive Distribution with Kernels" (2018).

[6] Teneggi et al. "How to Trust Your Diffusion Model: A Convex Optimization Approach to Conformal Risk Control" (2023).

[7] Belhasin et al. "Principal Uncertainty Quantification with Spatial Correlation for Image Restoration Problems" (2023).

[8] Fischer et al. "Subgroup-Specific Risk-Controlled Dose Estimation in Radiotherapy" (2024).

---

> ### Author Response · Authors · 2025-11-21
>
> We thank the reviewer for their constructive comments and thoughtful feedback. We are especially grateful for the in-depth and helpful comments on our methodology section. We provide answers to your specific questions and remarks below.
>
> **1.  Conformal Risk Control vs. Risk-Controlling Prediction Sets**
>
> We are grateful for the helpful feedback on our equations. We have included a more detailed explanation of our problem overview in section 3.1.
>
> Additionally, we recognized that we had previously interchanged the original Eq. 2 with the procedure described in Eq. 4 [2]. Our method actually follows Eq. 4 in [2], and we have now updated the corresponding citations and equation to reflect this correction. We also agree with your comment that the former Eq. 2 required the entire input image to be contained at once in a prediction set, rather than the pixels themselves. This has also been updated.
>
> We agree that our binary search algorithm does not rigorously achieve the stronger coverage notion, that methods in [3] do. We’ve updated eq. 2, to reflect this.
>
> **2. Multiple quantile estimation and quantile crossing**
>
> Thank you for this comment. We’ve included an additional table 3b that shows the quantile crossing occurrences for the predicted and lower bounds of Im2Im and QUTCC. Please note that both Im2Im and QUTCC exhibit some quantile-crossing violations, but the total occurrences are very small (most images do not exhibit them). In images with quantile crossing, the violations mainly occur in background areas, and the magnitude of this violation (the difference between the lower and upper quantiles) is usually in the 1e-3 to 1e-5 range.
>
> Nevertheless, we agree that these quantile crossings shouldn’t occur at all. In the related works section 2.2, we cite work that demonstrates that simultaneous quantile regression greatly mitigates this issue: “ In particular, estimating the quantiles jointly greatly alleviates the undesired phenomena of crossing quantiles” [4, 5]. Thus, we believe these effects are minimal.
> There are a few methods to further address this [8], like post-processing and applying constraints on the quantile predictions, but in the interest of time, we have not implemented this for the rebuttal.
>
> **3. Definition of the loss function**
>
> Thank you for this comment. We have revised Eq. (4) to explicitly formulate the loss as an expectation over the underlying data distribution, as well as over quantile levels sampled uniformly from [0,1].
>
> **4. Estimating the conditional distribution**
>
> We are grateful for the feedback. We tightened up the language around the true quantile function and the empirical quantile function in section 3.4.1 to avoid any confusion.
>
> To generate a PDF, we can evaluate multiple values of α=[0.1,0.2,0.3,… ]. The calibrated bounds will achieve coverage levels of [90% coverage, 80% coverage, 70% coverage, …]. If we then query all these quantiles together, in the resulting PDF, there is a coverage guarantee between the complementary bounds that x% of pixels will be between those bounds in the PDF.
>
> Thank you for highlighting existing methods that also provide estimates of distributions. While we ran out of time in the current rebuttal, we are considering adding a section to our related work specifically to discuss existing Bayesian approaches that estimate these distributions.
>
> **5. Experimental results**
>
> See main comment.
>
> Due to time constraints, we were unable to compare to [6, 7]. [7] derives uncertainty intervals around principal components, while our method looks at pixelwise uncertainty intervals. We are unsure that this would be a fair apples to apples comparison. [6] looks at the incorporation of uncertainty estimates into diffusion models. To make a fair comparison, we would either need to retrain our models on datasets compatible with diffusion models or extensively retrain or fine-tune the diffusion model on our existing datasets.
> The quantile embeddings enable QUTCC to predict the full range of quantiles, whereas Im2Im-Deep can only predict two fixed quantiles. If the question is why we cannot simply extend Im2Im-Deep to predict more fixed quantile bounds: Im2Im-Deep’s final layer is designed to produce only two fixed quantile losses (upper and lower bounds), which are added to the mean prediction and backpropagated through the network. We suspect that adding multiple fixed quantile losses in this final layer would degrade performance for both the bounds and the mean prediction.
>
>
> **continued down below**

---

> > ### Author Response · Authors · 2025-11-21
> >
> > **7. “The emoji in the title seems unrelated to the contribution of the paper? I would consider removing unless it contributes to the submission.”**
> >
> > We had hoped that our method, QUTCC, when spoken, would be pronounced phonetically like ‘cutesy.’ We’ve chosen to keep the emoji for this effect, but we’re open to feedback if there are strong opinions.”
> >
> > **8. “Typo in Eq. (8)”**
> >
> > This has been updated accordingly in the paper.
> >
> > **9. “In its current formulation, does Algorithm 1 always terminate? Since the step size does not change, isn't there a chance the thresholds will start oscillating?”**
> >
> > Thank you for this comment. Algorithm 1 is intended as pseudocode, and we have safeguards to ensure its termination. Specifically, a tolerance variable tracks the last valid quantile that satisfies the risk criterion; if subsequent quantiles fall within this tolerance, the algorithm terminates. Additionally, we observed that the algorithm typically converges within 20 iterations. We also have a variable that determines the number of iterations you run your calibration for and we have set this to be 20 in our code.
> >
> > **10. “The claim that no existing methods have addressed minimal interval length or conditional coverage in imaging inverse problems is too strong as there exist prior works that have proposed ways to tackle parts of these issues [6,8].”**
> >
> > Thank you for bringing these works to our attention. We have cited these in section 2.3 of our related work.
> >
> > **11. “The claim that prior works use linear scaling of the interval bounds might also be too broad, as several works have used additive parametrizations that do not scale intervals bounds multiplicatively, or different strategies such as masking.”**
> >
> > Thank you for bringing this to our attention! We are happy to add references to these works. Do you mind providing the relevant citations?
> >
> > Thank you again for your review. We hope we have addressed your questions and concerns. If you have any further questions, please do let us know.
> >
> > [1] Angelopoulos et al. "Image-to-Image Regression with Distribution-Free Uncertainty Quantification and Applications in Imaging" (2022).
> > [2] Angelopoulos et al. "Conformal Risk Control", (ICLR 2024).
> > [3] Angelopoulos et al. "Learn then Test: Calibrating Predictive Algorithms to Achieve Risk Control", (2022).
> > [4] Tagasovska, Natasa, and David Lopez-Paz. "Single-model uncertainties for deep learning." Advances in neural information processing systems 32 (2019).
> > [5] Takeuchi, Ichiro, et al. "Nonparametric quantile estimation." (2006).
> > [6] Teneggi et al. "How to Trust Your Diffusion Model: A Convex Optimization Approach to Conformal Risk Control" (2023).
> > [7] Belhasin et al. "Principal Uncertainty Quantification with Spatial Correlation for Image Restoration Problems" (2023).
> > [8] Dai, Sheng, Timo Kuosmanen, and Xun Zhou. "Non-crossing convex quantile regression." Economics letters 233 (2023): 111396.

---

> > > ### Comment · Reviewer_ak8f · 2025-11-21
> > > **Thank you for your response!**
> > >
> > > I sincerely thank the authors for their careful consideration of all reviewers' comments, and for their updates to the submission---presentation of the method has improved significantly.
> > >
> > > ---
> > >
> > > For an example reference of works not using linear scaling of uncertainty intervals, see Kutiel et al. "Conformal Prediction Masks: Visualizing Uncertainty in Medical Imaging", 2023.
> > >
> > > ---
> > >
> > > I updated my score accordingly. My score reflects:
> > >
> > > 1. Limited comparison with existing methods. I do think the authors, given rebuttal time limitations, have tried to address all reviewers comments to the best of their abilities. The updated tables do provide more evidence in support of the proposed method (line plots would be easier to read). However, if the main claim of the submission remains that the proposed method reduces interval length, it is important to include some method(s) beyond im2im (e.g., K-RCPS can be applied to quantile regressors as well, not diffusion models only). I understand comparisons might not be apples to apples, but they will help readers place the contribution of the submission better within the broader literature on uncertainty quantification for imaging inverse problems.
> > >
> > > 2. Presentation of main algorithm. It should be possible for a reader to reproduce the method by reading the main content of the paper. From the authors' response, there are some important implementation details missing from the current version of the submission. In my opinion, these are not minor details for publication.
> > >
> > > ---
> > >
> > > Overall, I think the submission puts forth a method that would be interesting to the community, but the method's contribution does not grant overhauling these aspects.
> > >
> > > P.S. I now understand the meaning of the emoji :)

---

> > > > ### Author Response · Authors · 2025-12-03
> > > >
> > > > Thank you for your thoughtful follow-up and for the updated score. We appreciate the time you have taken to assess the updated manuscript, as well as your suggestions.
> > > > Given the limited rebuttal period, we were unable to incorporate a new comparison model such as K-RCPS applied to quantile regressors, however this is something we are actively working on. We are also actively working on tightening up the presentation of our main algorithm so as to make it more reproducible.

---

### Official Review · Reviewer_gmqs · 2025-10-31

**Soundness:** 2
**Presentation:** 4
**Contribution:** 2
**Rating:** 2
**Confidence:** 5

**Summary:**

The authors propose a quantile embedding to enable nonlinear and non-uniform scaling of quantile predictions, thereby allowing for tighter pixel-based conformal interval lengths. The authors demonstrate their method on inverse problems using a U-Net, are able to estimate the full conditional distribution of quantiles for each image, and estimate the conditional CDF.

**Strengths:**

1. The manuscript is approachable and well written. The figures are clear.
2. The quantile embedding for image-to-image regression tasks is novel.
3. Estimating the conditional distribution and being able to mitigate quantile crossing offer practical utility.

**Weaknesses:**

1. The performance improvements compared to Im2Im-Deep (Table 1 and Table 2) are not entirely convincing and are marginal at best.
2. The crux of the method is the quantile embedding. However, the paper does not compare Im2Im-Deep on an equal playing field due to the different formulations (Im2Im-Deep uses symmetric, while QUTCC uses asymmetric). Specifically, it is unclear whether the performance gains are due to the asymmetric formulation, the quantile embedding, or both. It would make sense that the asymmetric formulation would perform better for higher pixel intensities, because NN models tend to accurately capture the more moderate/prevalent intensities well at the expense of less prevalent intensities. An apples-to-apples comparison between Im2Im-Deep with an asymmetric formulation would be beneficial to clarify this issue. I would be more than happy to increase my score with greater clarity and an additional baseline.
3. The manuscript would benefit from a more rigorous statement of the conformal guarantee.
4. The generation of the conformalized PDF is interesting, but it can be viewed as repeatedly applying the main calibration procedure at different $\alpha$ levels (brute force) rather than a fundamentally new technique.

**Questions:**

1. Why place a quantile embedding after each layer instead of picking a particular layer? An ablation study would be interesting to see.
2. In the supplementary, QUTCC requires more training epochs. Does the QUTCC training regime (sampling q randomly) require more careful hyperparameter tuning compared to baselines?
3. The conditional PDF is estimated using finite differences. How sensitive is the shape of the resulting PDF to the number and spacing of the quantile queries used to build the CDF?

---

> ### Author Response · Authors · 2025-11-21
>
> We thank the reviewer for their constructive comments and thoughtful feedback. We provide answers to your specific questions and remarks below.
>
> **1. “Why place a quantile embedding after each layer instead of picking a particular layer? An ablation study would be interesting to see.”**
>
> This is an excellent point. We have included the results of our quantile embedding ablation study in Section A.3.3. In this study, we compared our current model against variants with quantile embeddings placed in different locations (e.g., decoder only, encoder only, etc.). For a target alpha calibration of 0.1, only four variants achieved risk below this threshold. Among these, only our current QUTCC model achieves the smallest mean interval length.
>
> **2. “In the supplementary, QUTCC requires more training epochs. Does the QUTCC training regime (sampling q randomly) require more careful hyperparameter tuning compared to baselines?”**
>
> Thank you for this question. We maintain consistent hyperparameters across both QUTCC and Im2im models without any model-specific tuning. For epoch selection, we perform a comprehensive calibration sweep across all training epochs and select the checkpoint that produces the tightest uncertainty intervals. The difference in required training epochs stems from the fundamental learning objectives of each model. QUTCC requires additional training time because it must learn to predict the full range of quantiles across the entire distribution, whereas Im2im focuses on learning just two fixed quantile bounds.
>
> **3. “The conditional PDF is estimated using finite differences. How sensitive is the shape of the resulting PDF to the number and spacing of the quantile queries used to build the CDF?”**
>
> We’ve included an additional section in our supplement A.5.2, where we discuss this in more detail. You are correct in that the PDF is affected by the spacing of the quantile queries. However, we show in supp fig. 13, that the number of quantile queries does not vastly change the shape of the PDF distribution, as long as the spacing between each query stays consistent.
>
> **4. “the paper does not compare Im2Im-Deep on an equal playing field due to the different formulations (Im2Im-Deep uses symmetric, while QUTCC uses asymmetric).”**
>
> See main comment.
>
> Thank you again for your review. We hope we have addressed your questions and concerns. If you have any further questions, please do let us know.

---

> ### Comment · Reviewer_gmqs · 2025-11-23
>
> Thank you for addressing all the concerns I brought up. These solidify the utility of QUTCC. I have an additional concern/question regarding the novelty of the quantile embedding.
>
> The manuscript currently presents this embedding as a primary novelty. Upon further inspection of the method and analysis of prior literature, the quantile embedding in QUTCC appears to be structurally and mathematically identical to Implicit Quantile Networks (IQN) (Dabney et al., 2018). IQN samples a quantile $\tau \sim U([0,1])$, embeds it into a vector, and combines this embedding with the network's features (often via element-wise multiplication) to predict the value of the return at that specific quantile. This is also applied to generative image modeling in Ostrovski et al, 2018.
>
> Could you please explicitly outline the mathematical differences (if any) between your embedding strategy and that of IQN?
> 1. If there are no major differences, the manuscript must acknowledge IQN as the foundational method for this technique. The novelty would then lie in the application to inverse problems and conformal inference, rather than the quantile embedding itself.
> 2. It doesn't seem like this line of work is acknowledged in the manuscript. Could you please outline the main differences (if any) between the quantile embedding setup of QUTCC and IQN? Please provide a statement in the main text.
>
> Citations:
> 1. Dabney, Will, et al. "Implicit quantile networks for distributional reinforcement learning." International conference on machine learning. PMLR, 2018.
> 2. Ostrovski, Georg, Will Dabney, and Rémi Munos. "Autoregressive quantile networks for generative modeling." International Conference on Machine Learning. PMLR, 2018.

---

> > ### Author Response · Authors · 2025-11-26
> >
> > **We thank the reviewer’s careful review of the relevant prior literature.**
> > ***
> >
> > While our model incorporates quantile embeddings, we do not position this as the primary novelty of our work. Our main contribution is an efficient method that provides statistically grounded, pixel-wise uncertainty estimates for image regression tasks once calibrated. As stated in the abstract, “We propose QUTCC, a quantile uncertainty training and calibration technique … to enable tighter uncertainty estimates,” and in lines 89–90 of the introduction, “Building on past work in multi-quantile estimation, QUTCC uses a single neural network to estimate a distribution of quantiles.” We fully recognize that simultaneous estimation of the quantile function, or Simultaneous Quantile Regression (SQR), is an active body of research work, and we have aimed to reflect this appropriately in **Section 2.2** of our related work.
> >
> > IQN employs “quantile regression to approximate the full quantile function for the state–action return distribution,” enabling the model to represent a wide range of return distributions. Learning the entire quantile function is a form of SQR, and approaches for doing so have been extensively studied [1, 2]. However, there are a few key distinctions between the method employed in [3, 4] and QUTCC.
> >
> > **1. The quantile losses are different:** [3, 4] predominantly use the Quantile Huber Loss, while QUTCC uses Pinball loss. While both are asymmetric, this loss distinction is important.
> >
> > **2. We use self-attention:** Although both [3, 4] and QUTCC condition on a quantile level, the way they embed this information into the network is different. AQN/IQN employs a quantile embedding, implemented as a small MLP or cosine feature mapping that transforms the scalar quantile level into a conditioning vector. This vector is then merged with the hidden representation of an autoregressive convolutional model, like PixelCNN. QUTCC uses a diffusion-style embedding in which the quantile level is mapped to a high-dimensional feature vector that modulates activations throughout a U-Net architecture with self-attention layers. This embedding is used at multiple resolutions, which we discuss in **A.2 Model Architecture**. We’ve also conducted an ablation study for how many quantile embeddings to use at the request of another reviewer. If interested, please refer to **section A.3.3**.
> >
> > **3. Different fundamental purposes between ACN/IQN/QUTCC:** IQN is designed for reinforcement learning purposes. AQN is designed for generative modeling: conditioning on a quantile level allows the model to represent and sample from the underlying data distribution via autoregressive decoding. Its quantile embedding is therefore a mechanism for guiding the generator toward different parts of the distribution. QUTCC, focuses on pixel-wise uncertainty quantification for image regression and inverse problems. By learning a continuous quantile function for each pixel, and pairing this with conformal calibration, QUTCC provides statistically valid pixelwise uncertainty intervals for reconstructed images.
> > ***
> >
> > We have updated **Section 2.2 (Related Work)** and **Section 3.2 (Network Architecture)** in **blue text** to clarify these connections and differences. **We appreciate the reviewer bringing this prior work to our attention. As our project is oriented toward the imaging community, we did not emphasize links to reinforcement learning or generative modeling as clearly as we should have.**
> >
> >
> > [1] Yufeng Liu and Yichao Wu. Simultaneous multiple non-crossing quantile regression estimation using kernel constraints. Journal of nonparametric statistics, 23(2):415–437, 2011.
> >
> > [2] Maxime Sangnier, Olivier Fercoq, and Florence d’Alché Buc. Joint quantile regression in vector-valued rkhss. Advances in Neural Information Processing Systems, 29, 2016.
> >
> > [3] Dabney, Will, et al. "Implicit quantile networks for distributional reinforcement learning." International conference on machine learning. PMLR, 2018.
> >
> > [4] Ostrovski, Georg, Will Dabney, and Rémi Munos. "Autoregressive quantile networks for generative modeling." International Conference on Machine Learning. PMLR, 2018.

---

### Official Review · Reviewer_y9oy · 2025-10-31

**Soundness:** 3
**Presentation:** 4
**Contribution:** 3
**Rating:** 2
**Confidence:** 3

**Summary:**

The authors propose quantile uncertainty training and conformal calibration (QUTCC), a method for training quantile regression neural networks to estimate the full quantile function (as opposed to a fixed set of quantiles, e.g., 0.05 and 0.95), and how to obtain conformalized prediction bounds using this approach. While the proposed method is general, the authors focus on its application for image-to-image regression. Extensive experiments are conducted across five imaging inverse problems and compared against an appropriate baseline (Im2Img-UQ). The authors claim that the proposed method consistently estimates narrower uncertainty intervals compared to the baseline, while achieving the desired level of risk.

**Strengths:**

Novelty and relevance: While the idea of adding conditioning mechanisms to enable a quantile regression network to learn the full quantile function is not novel (see [1, 2]), its application for uncertainty prediction in inverse imaging problems is novel, relevant, and timely.

Extensive experiments: The authors conduct a thorough evaluation of their proposed method on five different datasets, selecting an appropriate baseline for comparison.

References:
[1] Ostrovski, Georg, Will Dabney, and Rémi Munos. "Autoregressive quantile networks for generative modeling." International Conference on Machine Learning. PMLR, 2018.
[2] Dabney, Will, et al. "Implicit quantile networks for distributional reinforcement learning." International conference on machine learning. PMLR, 2018.

**Weaknesses:**

Major weaknesses:
Insufficient improvements over baseline: My main criticism of this work is its minor (possibly zero?) improvements over the baseline Im2Im-UQ. The results shown in Table 1 and Figure 2 suggest nearly identical performance. Even in Figure 4, the selected outputs of the baseline and proposed methods are extremely similar. Consequently, it is difficult to justify that predicting the full quantile function versus 2 fixed quantile values (0.05 and 0.95) offers any real benefit.

Minor weaknesses:
Alternate architectures: The authors chose to fix the UNet architecture (ResNet-18 backbone) used for the experiments to ensure a fair comparison between the proposed method (QUTCC) and the baseline method (Im2Im-UQ). While I do think this is completely appropriate, it would be nice to see this comparison for a variety of architectures. Nevertheless, this is a very minor point, and I don’t believe it would substantially alter the main results of this work.

**Questions:**

Are there any other benefits of estimating the full quantile function over a few fixed quantile values apart from obtaining tighter prediction intervals?

Are there any differences in the model’s internal representations between using QUTCC vs. fixed quantiles?

---

> ### Author Response · Authors · 2025-11-21
>
> We thank the reviewer for their constructive comments and thoughtful feedback. We provide answers to your specific questions and remarks below.
>
> **1. “Are there any other benefits of estimating the full quantile function over a few fixed quantile values apart from obtaining tighter prediction intervals?”**
>
> Thank you for this question and we’re happy to clear this up. In our related work, we note that estimating the full quantile function can mitigate quantile crossing while also introducing a useful regularization effect for predictive performance [1, 2, 3]. In our approach, predicting the entire range of quantiles further enables us to recover a pixelwise probability density function (PDF). This PDF provides conditional distributions with statistical coverage guarantees and is fully data-driven, requiring no additional assumptions about the underlying data distribution. In Fig. 5, we show that at different pixel locations, the PDF predicts different types of distributions. Additionally, we’ve added Fig. 12 in the supplement, which demonstrates that our PDF predictions can capture different types of noise distribution (Gaussian, Poisson).
>
> **2. “Are there any differences in the model’s internal representations between using QUTCC vs. fixed quantiles?”**
>
> We appreciate the opportunity to clarify this point. Our model architecture differs from the original Im2Im, which uses a standard UNet. To ensure that our observed performance improvements stem from our uncertainty quantification method rather than architectural changes, we also compare QUTCC against Im2Im-Deep, a model with the same architecture as QUTCC but without quantile embeddings. The quantile embeddings enable QUTCC to predict the full range of quantiles, whereas Im2Im-Deep can only predict two fixed quantiles. Predicting the full range of quantiles provides several benefits (see our response to point 1). If the question is why we cannot simply extend Im2Im-Deep to predict more fixed quantile bounds to construct a PDF: Im2Im-Deep’s final layer is designed to produce only two fixed quantile losses (upper and lower bounds), which are added to the mean prediction and backpropagated through the network. We suspect that adding multiple fixed quantile losses in this final layer would degrade performance for both the bounds and the mean prediction. By contrast, using quantile embeddings and querying random quantiles during training allows QUTCC to efficiently learn the full quantile function without compromising predictive accuracy.
>
> **3. “Insufficient improvements over baseline: My main criticism of this work is its minor (possibly zero?) improvements over the baseline Im2Im-UQ. The results shown in Table 1 and Figure 2 suggest nearly identical performance.”**
>
> See main comment
>
> Thank you again for your review. We hope we have addressed your questions and concerns. If you have any further questions, please do let us know.
>
> [1] Filipe Rodrigues and Francisco C Pereira. Beyond expectation: Deep joint mean and quantile regression for
> spatiotemporal problems. IEEE transactions on neural networks and learning systems, 31(12):5377–5389,
> 2020
>
> [2] Maxime Sangnier, Olivier Fercoq, and Florence d’Alché Buc. Joint quantile regression in vector-valued rkhss.
> Advances in Neural Information Processing Systems, 29, 2016.
>
> [3] Yufeng Liu and Yichao Wu. Simultaneous multiple non-crossing quantile regression estimation using kernel
> constraints. Journal of nonparametric statistics, 23(2):415–437, 2011.

---

### Official Review · Reviewer_ZRdo · 2025-11-03

**Soundness:** 4
**Presentation:** 3
**Contribution:** 3
**Rating:** 8
**Confidence:** 5

**Summary:**

The paper introduces QUTCC, a framework for uncertainty quantification in imaging inverse problems. It proposes a quantile-conditioned U-Net that predicts conditional quantiles as a function of a sampled quantile level, trained using a pinball loss akin to quantile regression. Similar to how diffusion models condition on timesteps, QUTCC conditions on quantiles, enabling the network to predict the full conditional distribution of image intensities. A nonlinear, non-uniform conformal calibration step then adjusts quantile bounds to guarantee statistical coverage while producing tighter uncertainty intervals. The model can be queried at test time to yield median predictions, pixel-wise confidence intervals, and even per-pixel probability density functions (PDFs). Across multiple imaging tasks—including denoising, MRI, and quantitative phase imaging—QUTCC achieves smaller, sharper uncertainty bounds and better hallucination detection compared to prior methods like Im2Im-UQ.

**Strengths:**

The paper presents a principled and rigorous approach to uncertainty estimation in imaging inverse problems. Its key contribution—the quantile-conditioned U-Net trained with a pinball loss and paired with nonlinear conformal calibration—represents a natural yet nontrivial extension of prior conformalized quantile regression methods such as Im2Im. The methodology is conceptually elegant and statistically grounded, providing a unified framework for pixel-wise uncertainty quantification, coverage guarantees, and conditional density estimation. The paper is clearly written, with informative figures that effectively illustrate both the conceptual flow and empirical results. Experimental validation across multiple imaging modalities and noise models (MRI, QPI, Gaussian, Poisson, and real noise) convincingly demonstrates robustness and generality, highlighting the method’s potential significance for reliable deep imaging.

**Weaknesses:**

While the paper is technically sound and well presented, several aspects could be improved to strengthen its contribution. First, the related work section omits an important body of literature on multi-modal prediction and multi-hypothesis uncertainty estimation using multi-head or mixture-based networks (e.g., Learning in an Uncertain World: Representing Ambiguity Through Multiple Hypotheses by Rupprecht et al., Hierarchical Uncertainty Exploration via Feedforward Posterior Trees by Nehme et al.). These works share conceptual overlap with simultaneous quantile prediction and discussing them in the related work section can better contextualize the paper’s novelty. Second, the interpretation of pixel-wise marginal distributions raises conceptual concerns: modeling uncertainty independently for each pixel neglects spatial correlations that are fundamental to imaging tasks. It remains unclear how these marginal PDFs can be practically leveraged beyond visualization, and Figure 5 does not convincingly demonstrate their utility. Finally, the quantitative results in Table 1, while consistent, show only marginal improvements over the baseline, and stronger ablations or uncertainty-coverage tradeoff analyses would help substantiate the claimed performance gains.

**Questions:**

Questions and suggestions for the authors:
1. Could the authors comment on the potential complementarity between multi-modal prediction frameworks (e.g., multi-head or mixture density networks) and quantile-based uncertainty estimation? It would be interesting to understand whether quantile conditioning could be integrated into architectures designed for multi-hypothesis prediction.
2. In Equation (8), shouldn’t the inequality be $\mathbb{P}(\\hat{\\mathbf{x}}\_{lower}(k)\\le\\mathbf{x}\\le\\hat{\\mathbf{x}}\_{upper}(k)) \\ge 1 - \\alpha$? Can you clarify this?
3. The calibration procedure in Algorithm 1 appears similar in spirit to the approach used in Im2Im. Could the proposed conformal calibration, in principle, have been implemented within that framework as well?
4. What practical insight or downstream utility is gained from the pixel-wise marginal distributions shown in Figure 5? As presented, these marginals appear difficult to interpret or use for actionable uncertainty quantification.
5. In Table 1, is the comparison between QUTCC around the same predictor? QUTCC reports median-based estimates, is Im2Im also doing that? or is it using mean prediction? Aligning the central statistic across methods is crucial and could affect the reported results.
6. How sensitive is the proposed model to the quantile sampling strategy used during training? For example, do uniform versus importance-weighted quantile samples impact calibration quality, coverage, or stability during optimization?

---

> ### Author Response · Authors · 2025-11-21
>
> We thank the reviewer for their constructive comments and thoughtful feedback. We provide answers to your specific questions and remarks below.
>
> **1. “Could the authors comment on the potential complementarity between multi-modal prediction frameworks (e.g., multi-head or mixture density networks) and quantile-based uncertainty estimation? It would be interesting to understand whether quantile conditioning could be integrated into architectures designed for multi-hypothesis prediction.”**
>
> Thank you for bringing this to our attention! We’ve added additional references and citations to multi-head and multi-hypothesis methods in section 2.3 to highlight the conceptual overlap. Additionally, we’ve included future work where we discuss incorporating quantile conditioning into multi-hypothesis prediction networks.
>
> **2. The typo in equation (8)**
>
> This is in fact a typo. We have corrected it.
>
> **3. “The calibration procedure in Algorithm 1 appears similar in spirit to the approach used in Im2Im. Could the proposed conformal calibration, in principle, have been implemented within that framework as well?”**
>
> Our core contribution is that we utilize quantile embeddings during training to learn the quantiles of the underlying distribution. Previous methods, like Im2Im and Conformalized Quantile Regression use a neural network that predicts a fixed lower and upper bound instead of the full conditional quantile distribution. This means that during conformal calibration, we can leverage a non-uniform scaling that’s a function of the network instead of updating a constant scaling to reach the desired coverage. Our approach builds off of the Im2Im calibration procedure and extends it to work with simultaneous quantile regression.
>
> **4. “What practical insight or downstream utility is gained from the pixel-wise marginal distributions shown in Figure 5? As presented, these marginals appear difficult to interpret or use for actionable uncertainty quantification.”**
>
> Thank you for bringing this point up. We included Figure 5 in the text to show that the PDF predictions in QUTCC are capable of predicting different types of distributions (rather than just purely Gaussian assumptions). To further illustrate this point, we’ve included section A.5.1 and Fig. 12 in the supplement to show that QUTCC can also capture different noise distributions- poisson and gaussian. This capability is valuable for downstream applications that require identifying or validating the underlying noise model.
>
> **5. “In Table 1, is the comparison between QUTCC around the same predictor? QUTCC reports median-based estimates, is Im2Im also doing that? or is it using mean prediction? Aligning the central statistic across methods is crucial and could affect the reported results.”**
>
> See main comment.
>
> **6. “How sensitive is the proposed model to the quantile sampling strategy used during training? For example, do uniform versus importance-weighted quantile samples impact calibration quality, coverage, or stability during optimization?”**
>
> In our method, we use uniform quantile sampling. Beyond predicting upper and lower confidence bounds, our goal is for the model to accurately estimate the intermediate quantiles as well, enabling reliable reconstruction of the full PDF. If someone only cares about predicting uncertainty bounds, an alternative strategy would be to use importance-weighted sampling that concentrates more quantiles near the extremes. This would allow the model to more effectively learn the behavior of the tails.
>
> **7. “Second, the interpretation of pixel-wise marginal distributions raises conceptual concerns: modeling uncertainty independently for each pixel neglects spatial correlations that are fundamental to imaging tasks. It remains unclear how these marginal PDFs can be practically leveraged beyond visualization, and Figure 5 does not convincingly demonstrate their utility.”**
>
> See response to point 4.
>
> Thank you again for your review. We hope we have addressed your questions and concerns. If you have any further questions, please do let us know.

---

### Author Response · Authors · 2025-11-21
**Main Rebuttal Response**

We were delighted to see four detailed and highly informative reviews of our paper. Thank you to all our reviewers for taking the time to read and provide feedback on our work.  We appreciate the reviewers’ recognition of our approach as a timely, relevant, and innovative method for uncertainty quantification in image regression tasks. We have made several revisions to the paper in response to the reviewer questions and feedback and have uploaded a revised draft with changes highlighted in red.

**There were a few main concerns that all reviewers brought up and we have detailed our changes below.**

**1. All reviewers noted that the gains demonstrated in Figure 1 and Figure 2 are insufficient to prove that QUTCC is outperforming the baseline method**

In response to this, we have added section A.4 and fig 6 in the supplementary, where we stratify the uncertainty interval sizes by pixel intensity. As highlighted in Fig. 3, the gains in interval estimation provided by QUTCC primarily arise in regions of higher pixel intensity, while regions of lower intensity (background) exhibit performance similar to Im2Im. This trend is further illustrated in Fig. 6, where interval lengths in background regions are comparable across methods, but in regions of high pixel intensity, QUTCC generally predicts shorter, more precise intervals. For Gaussian, Poisson, and Real Noise tasks, the difference in interval length becomes more pronounced the larger the pixel intensity.

In response to reviewer feedback regarding the fairness of comparisons between Im2Im and QUTCC, we have added two additional baselines to Fig.6: Im2Im-Median and Im2Im-Asymm. Im2Im-Median corresponds to the original Im2Im model trained with a median loss (quantile = 0.5) instead of MSE, ensuring that both QUTCC and Im2Im-Median optimize losses centered on the same statistic. Im2Im-Asymm implements asymmetric calibration by applying different scaling factors to the upper and lower bounds, addressing concerns that QUTCC's gains could arise solely from its asymmetric calibration of the bounds. Even when compared to Im2Im-Asymm, QUTCC produces shorter interval lengths, demonstrating the effectiveness of our approach.

**2. Reviewers were curious about why predicting multiple quantiles is important**

A key contribution of our work is that we leverage a single network to simultaneously estimate multiple quantiles, which leads to two notable advantages. First, this eliminates the need for a constant, linear scaling term during conformal calibration, as is required in Im2Im. Second, by repeatedly querying these quantiles, we can construct a pixelwise probability density function. This approach removes the need to make assumptions about the underlying distribution. Moreover, as shown in Fig. 12 of the supplementary material, it allows us to infer different types of noise distributions directly from the prediction

**We hope this addresses the main concerns of the reviewers. We will respond to the remainder of the comments individually. Thank you.**

---

### Meta-Review · Area_Chair_zi9p · 2026-01-03

**Summary:**

In the initial round of reviews, one reviewer evaluated the paper as above the acceptance threshold (score 8) and three reviewers evaluated it as below the threshold (scores 2,2,2). The main concerns were: (1) unclear applicability of pixel-wise uncertainty prediction to downstream tasks (a comment relevant to all methods in this category, not only to this paper); (2) very minor improvement over the baseline method, well below 1 standard deviation; (3) potentially unfair comparison to the baseline due to the use of asymmetric vs. symmetric loss, and centering around the posterior median rather than the posterior mean; (4) non-rigorous mathematical presentation; (5) incorrect framing as the first method to use minimal interval length or conditional coverage or nonlinear scaling of the interval bounds.

The authors’ response addressed most of the reviewer’s concerns. In particular, comparisons to the baseline were added with the same settings, and discussions on relevant previous work were added. Two reviewers engaged in further discussion, in which an additional point was raised: (6) the approach is highly similar to Implicit Quantile Networks (IQN) (Dabney et al., 2018), a method that was initially introduced in the context of RL. The authors acknowledge the approach is similar, but pointed there are differences in the loss, the network architecture, and goal (imaging vs. RL). No validation of the advantage of the chosen loss over that used in IQN was provided.

The AC views the paper as an interesting contribution. However, the AC agrees with the reviewers that the advantage over the baseline is very minor in terms of interval length. Even though there is a significant advantage for QUTCC at high intensity values for Gaussian and Poisson noise, as highlighted by the authors, the same figure shows that QUTCC underperforms at high intensity values in the MRI experiment. Therefore, overall, there is no consistent significant advantage for QUTCC across tasks, even if restricting attention only to high intensity values. As for the ability to extract the entire quantile function, this by itself is also not new, as pointed out by Reviewer gmqs, even if originally done in a different context, and the effect of the difference in the losses has not been illustrated. Therefore, the AC views the paper as not ready for publication in this ICLR.

**Reviewer Concerns:**

The rebuttal provided point-by-point answers to all the questions. However, as pointed out by reviewers ak8f and gmqs, the paper lacks a more thorough positioning and comparison to previous approaches, and the benefit over the baseline in terms of quantile lengths seems very minor.

**Reviewer Scores:**

**ZRdo: score 8.**

This is also the original score.

**y9oy: score 2.**

This is also the original score. The AC believes that the authors’ answer regarding the benefit of the method over outputting several fixed quantiles would not convince the reviewer. The authors state that learning the entire quantile function has a regularization effect, but having a single network output several quantiles simultaneously may also have a regularization effect. Given that QUTCC barely improves upon a network that outputs two quantiles, maybe this tiny advantage would completely vanish against a network that outputs more than two quantiles.

**gmqs: score 4.**

The original score was 2. The reviewer seems to have been satisfied with the authors’ response to the initial review, however in the discussion the reviewer brought up the similarity to IQN, which seems to require a more thorough treatment in the manuscript.

**ak8f: score 4**

The original score was 2. Some of the reviewer’s concerns were addressed. However, the reviewer indicated that the paper still lacks comparisons with some existing methods.

---

### Decision · Program_Chairs · 2026-01-26

Reject